# Impacts of Max-Stable Process Areal Exceedance Calculations to Study Area Sampling Density, Surface Network Precipitation Gage Extent and Density, and Model Fitting Method

**Brian Skahill** [1],*[ID], **Cole Haden Smith** [2], **Brook T. Russell** [3] **and John F. England** [2]

[1] U.S. Army Corps of Engineers, Engineer Research and Development Center, Coastal and Hydraulics Laboratory, Vicksburg, MS 39180, USA
[2] U.S. Army Corps of Engineers, Institute for Water Resources, Risk Management Center, Lakewood, CO 80228, USA; cole.h.smith@usace.army.mil (C.H.S.); john.f.england@usace.army.mil (J.F.E.)
[3] School of Mathematical and Statistical Sciences, Clemson University, Clemson, SC 29634, USA; brookr@clemson.edu
* Correspondence: brian.e.skahill@usace.army.mil

**Abstract:** Max-stable process (MSP) models can be fit to data collected over a spatial domain to estimate areal-based exceedances while accounting for spatial dependence in extremes. They have theoretical grounding within the framework of extreme value theory (EVT). In this work, we fit MSP models to three-day duration cool season precipitation maxima in the Willamette River Basin (WRB) of Oregon and to 48 h mid-latitude cyclone precipitation annual maxima in the Upper Trinity River Basin (TRB) of Texas. In total, 14 MSP models were fit (seven based on the WRB data and seven based on the TRB data). These MSP model fits were developed and applied to explore how user choices of study area sampling density, gage extent, and model fitting method impact areal precipitation-frequency calculations. The impacts of gage density were also evaluated. The development of each MSP involved the application of a recently introduced trend surface modeling methodology. Significant reductions in computing times were achieved, with little loss in accuracy, applying random sample subsets rather than the entire grid when calculating areal exceedances for the Cougar dam study area in the WRB. Explorations of gage extent revealed poor consistency among the TRB MSPs with modeling the generalized extreme value (GEV) marginal distribution scale parameter. The gauge density study revealed the robustness of the trend surface modeling methodology. Regardless of the fitting method, the final GEV shape parameter estimates for all fourteen MSPs were greater than their prescribed initial values which were obtained from spatial GEV fits that assumed independence among the extremes. When two MSP models only differed by their selected fitting method, notable differences were observed with their dependence and trend surface parameter estimates and resulting areal exceedances calculations.

**Keywords:** max-stable process; spatial dependence; trend surface; areal exceedance; extreme value theory; extreme precipitation; Willamette River Basin; Trinity River Basin

## 1. Introduction

Extreme precipitation frequency areal estimates over watersheds are a key component in estimating flood hazards and hydrologic risk [1,2]. Flood frequency estimation within the US Army Corps of Engineers (USACE) dam and levee safety program involves combining limited at-site flood data with temporal information on historic and paleofloods, spatial information on areal precipitation frequency, and causal information based on the hydrologic modeling of rainfall-frequency events to enhance and expedite flood hazard assessments [3–6]. Precipitation frequency estimates for dam and levee safety span a range of annual exceedance probabilities (AEPs) typically from $10^{-2}$ to $10^{-7}$ and require credible extrapolation and robust uncertainty estimates [7]. Current practice in precipitation frequency

for dam safety uses GEV/L-moments and areal reduction factors [7]. Extreme precipitation frequency estimates are also a necessary component for other risk assessments [8,9].

Advances from the field of extreme value theory (EVT) have demonstrated the capacity to efficiently, flexibly, and credibly model spatial extremes of pointwise maxima using a max-stable process (MSP) [10–14], the infinite-dimensional analog of multivariate extreme value distribution. The application of an MSP model enables the direct estimation of areal-based exceedances within an EVT-based framework. MSP models explicitly account for the spatial dependence of the extreme data [13,14]. They do not depend upon the subjective assumptions associated with a Regional Precipitation Frequency Analysis (RFA) [15], for example, the definition of homogeneous subareas and the need to convert point estimates into areal average depths using uncertain empirical regional depth-area reduction factors [16,17]. They also do not share a disadvantage of an RFA which does not construct an explicit spatial model for the marginal parameters [10]. With their application, one can not only compute pointwise return level maps, but also more complex areal-based assessments of risk such as $\Pr\{\int_{\mathcal{B}} Y(x)dx > z_{crit}\}$, where $Y(x)$, $\mathcal{B}$, and $z_{crit}$ denote the joint distribution, any arbitrary area within the analysis domain, for example, a sub-basin of interest, and a critical quantity greater than zero, respectively [13,18].

Several studies have fit MSPs to block maxima in order to model precipitation extremes [19–32]. This study differs from these previous studies in that it focused on four unexplored issues of practical importance for the calculation of areal exceedances when applying an MSP model for the analysis of extreme precipitation.

With an MSP model, areal exceedance estimates are obtained by simulating multiple independent copies of the fitted process over an area of interest [12,13,33], using its fitted parameter estimates or random samples either gleaned from model calibration [10,13,18,26,34] or application of the bootstrap method [35,36]. In practice, estimating areal exceedances in this manner can be computationally intensive, particularly when large areas or rare AEPs are of interest. This study examined the impact of using a random sample subset of the entire grid composing an area of interest on the calculation of areal exceedances when using an MSP model.

The spatial structure in marginal extreme precipitation behavior at all locations throughout a study region can be effectively modeled at various scales [37]. This study investigated the impact the spatial precipitation gage extent selected for an MSP model deployment had on the calculation of areal exceedances. We also explored how well a precipitation gauge network observed a spatial process of extreme precipitation.

Gage density and configuration are not uniformly distributed throughout most basins, and these characteristics of a surface network impact the estimation of spatially varying precipitation [38,39]. This study explored the degree of agreement among areal precipitation-frequency estimates calculated from distinct MSPs developed using a surface network of rainfall gages of varying relative density.

MSP model fitting can be thought of as a two-step procedure involving trend surface and simple MSP model selection, with each step assuming independence among the extremes and fixed unit Fréchet margins, respectively [13,33]. Trend surfaces are functions of geographical and/or climatological covariates that influence regional precipitation extremes to model the spatial variation of the location, scale, and shape parameters of the known generalized extreme value (GEV) marginal distributions [13,14,37]. Trend surface parameterizations can potentially complicate dependence parameter estimation [13,34,40]. In this study, linear trend surfaces for the marginal parameters were estimated by applying the methods described by Love et al. [37], which leveraged theory from spatial extremes and recent advances for regularizing general linear models [41–43]. Despite the application of a novel and effective trend surface modeling approach (Love et al. [37]; Ribatet [13]), this study also examined the impact the selected general MSP model fitting method had on areal exceedance estimation.

These four issues, relevant to the practical application of MSPs, were addressed using a series of MSP model fits that were estimated based on the cool season (October to April)

three-day duration and 48 h mid-latitude cyclone precipitation annual maxima in the Willamette River Basin of Oregon and the Trinity River Basin in Texas, respectively.

## 2. Materials and Methods

### 2.1. Study Areas

The 29,728 square kilometer Willamette River basin (WRB) located in northwestern Oregon is a major tributary of the Columbia River whose 301 km long main stem, the Willamette River, flows northward between the Coastal and Cascade Ranges (Figure 1). The U.S. Army Corps of Engineers (USACE) Portland District operates thirteen dams in the WRB. Within the WRB, areal exceedances were calculated for the 536 square kilometer Cougar Dam project study area (Figure 1).

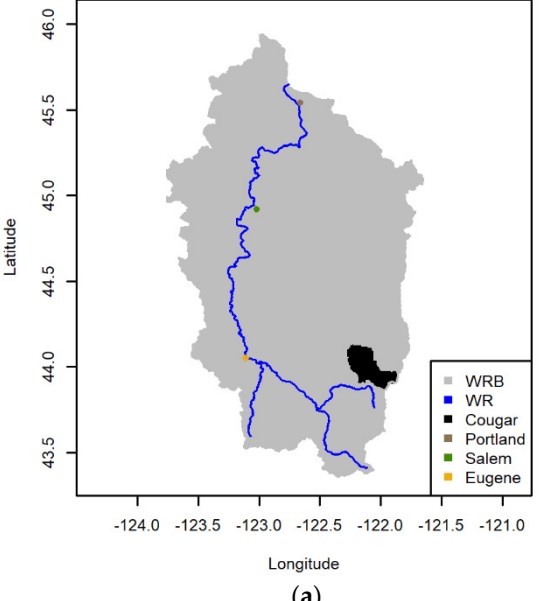
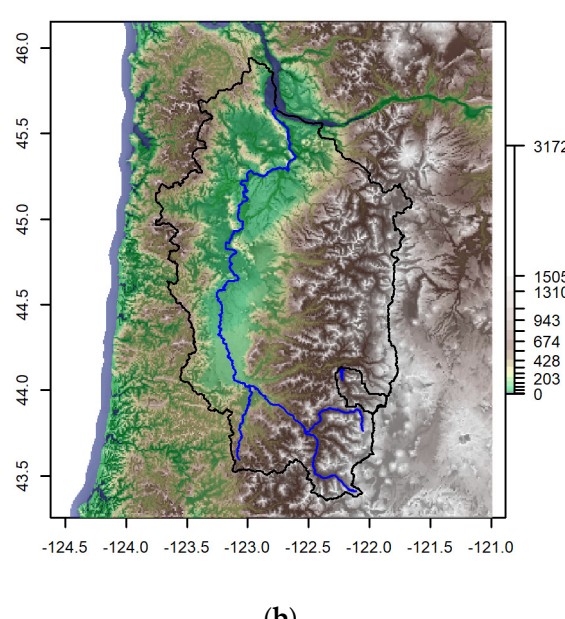

**(a)**　　　　　　　　　　　　　　　　**(b)**

**Figure 1.** (**a**) The 29,728 square kilometer Willamette River Basin (WRB), 301 km long Willamette River (WR), 536 square kilometer Cougar dam safety project contributing drainage area, and locations for the cities of Portland, Salem, and Eugene. (**b**) Digital elevation model (elevations in meters), including boundaries for the WRB and Cougar dam safety project contributing drainage area. Areal precipitation-frequency estimates were calculated for the 536 square kilometer Cougar Dam project study area. For each plot, the horizontal axis is in degrees longitude and the vertical axis is in degrees latitude.

Strong atmospheric rivers typically impact the West Coast during the cool season (November–April) [44]. Ralph and Dettinger [44] determined that an atmospheric river (AR) struck the West Coast during all 3-day duration rainfall events from 1997–2005 west of 115° W when totals from at least one weather station exceeded 300 mm. The Northwest Coast experiences the most ARs, with the greatest frequencies occurring between 43–46° N [45]. Extreme precipitation events over the western Cascades are mostly attributed to anticyclonic wave breaking (AWB) ARs which generally occur at latitudes greater than 43° N and strike the West Coast predominantly in a westerly direction [45]. Cyclonic wave breaking (CWB) ARs impinge the West Coast in a southwesterly direction and to a lesser degree account for extreme precipitation in the western Cascades [45].

Extreme precipitation areal exceedances were calculated for a 15,662 square kilometer project study area composed of the Upper West Fork Trinity (12030101; 5102 square kilometers), Lower West Fork Trinity (12030102; 3911 square kilometers), Elm Fork Trinity (12030103; 4766 square kilometers), and Denton (12030104; 1883 square kilometers) United States Geological Survey (USGS) 8-digit Hydrologic Unit Code (HUC) sub-basins located in the upper Trinity River Basin of Texas (Figure 2). Five high-hazard dams located in the Upper

Trinity River Basin provide flood protection for the city of Dallas, Texas. The areal exceedances were calculated for storms classified as 48 h extratropical cyclones. Precipitation from extratropical cyclones causes most major floods in large river basins throughout the conterminous United States [46]. In the southern United States, extratropical cyclones are common in late winter and early spring (December through March) [46–48]. While their intensity and frequency vary considerably [48], they can produce substantial rainfall [46].

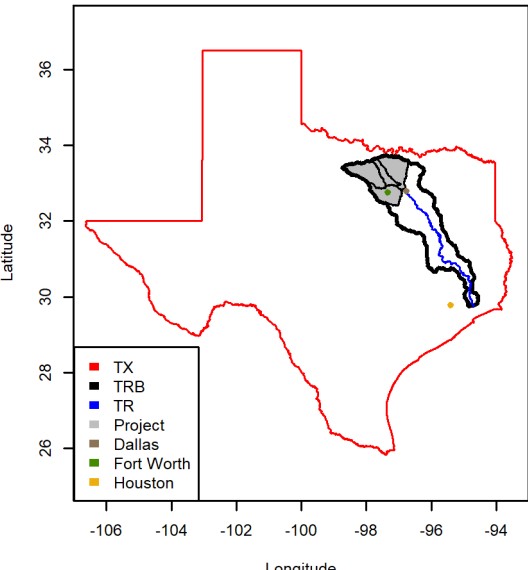

**Figure 2.** The state of Texas (TX), 47,000 square kilometre Trinity River Basin (TRB), 1140 kilometres long Trinity River (TR), 15,662 square kilometre project study area composed of the Upper West Fork Trinity (12030101; 5102 square kilometres), Lower West Fork Trinity (12030102; 3911 square kilometres), Elm Fork Trinity (12030103; 4766 square kilometres), and Denton (12030104; 1883 square kilometres) United States Geological Survey 8-digit Hydrologic Unit Code sub-basins located in the upper Trinity River Basin of Texas, and locations for the cities of Dallas, Forth Worth, and Houston. Areal precipitation-frequency estimates were calculated for the 15,662 square kilometre project study area. The horizontal axis is in degrees longitude and the vertical axis is in degrees latitude.

### 2.2. Block Maxima Precipitation Datasets

For the WRB, this study used subsets of a cool season (October to April) 3-day duration block maximum precipitation dataset that was compiled for the entire Columbia River Basin [49]. A detailed description of the original extreme precipitation data collection and processing procedure was provided in Appendix B of Skahill et al. [49]. The original seasonal maximum precipitation data collection ranged from the years 1867–2018 (152 years).

The first subset included the 140 gages from Skahill et al. [49] with at least 20 seasonal maxima observations [50] that were located within the WRB and a 20-km buffer of the WRB watershed boundary (Figure 3). A second subset involved the 286 precipitation gages from Skahill et al. [49] with at least 20 seasonal maxima observations whose footprint intersected with the 295 precipitation gages that were used for a relatively recent L-moments RFA that was performed for the WRB [51] (Figure 3). The third precipitation gage dataset was a subset of the first and it considered the 26 precipitation gages located within the 1-degree-by-1-degree box with north, south, east, and west extents of 44.75° N, 43.75° N, −121.8° W, and −122.8° W, respectively (Figure 3). This study also considered three additional randomly sampled subsets of the first, of sizes 35, 70, and 105 precipitation gages, respectively (Figure 3). The total number of seasonal maxima observations for the 140, 286, 26, 35, 70, and 105 precipitation gages shown in Figure 3a–f were 7274, 14,747, 1260, 1973, 3518, and 5632, respectively.

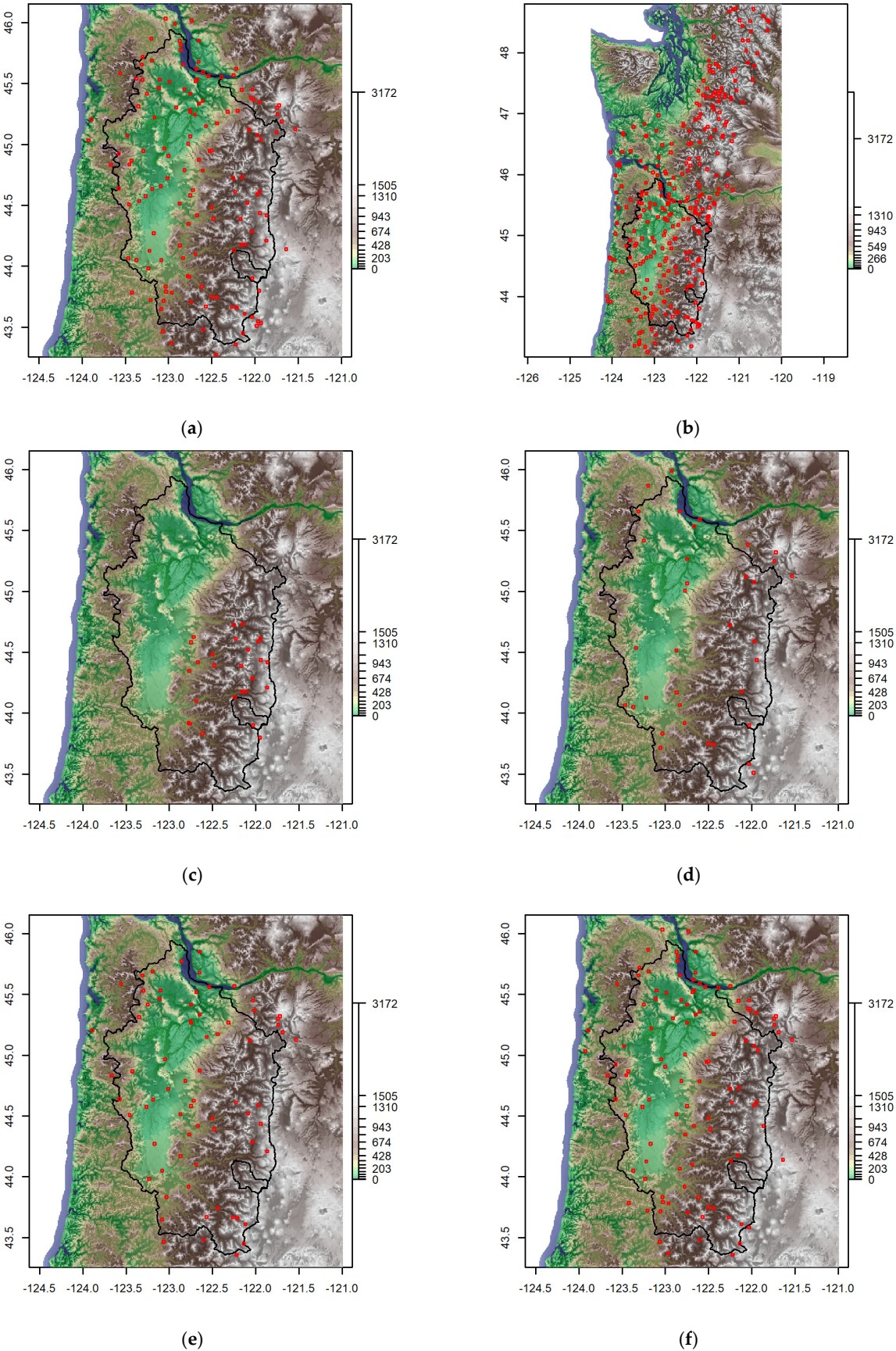

**Figure 3.** Precipitation gage subsets, of the cool season (October to April) 3-day duration block maximum precipitation dataset that was compiled for the entire Columbia River Basin [49], of sizes

(**a**) 140, (**b**) 286, (**c**) 26, (**d**) 35, (**e**) 70, and (**f**) 105. For each plot, the background map is a digital elevation model (elevations in meters). These precipitation gage datasets were prepared and used to develop and apply distinct MSPs to model cool season (October to April) three-day duration precipitation maxima in the Willamette River Basin of Oregon. For each plot, the horizontal axis is in degrees longitude and the vertical axis is in degrees latitude.

For the TRB, this study used subsets of a 48 h duration mid-latitude cyclone (MLC) annual maximum (July to June) precipitation dataset that was compiled for the 8-degree-by-6-degree rectangular region with north, south, east, and west extents of approximately 29° N, 35° N, −101° W, and −93° W [52]. A detailed description of the original extreme precipitation data collection and processing procedure was provided by Martin et al. [52] and Martin et al. [53]. The original dataset included 931 precipitation gages.

Four subsets of the original 48 h duration MLC annual maximum precipitation dataset included the 85, 151, 360, and 610 gages located within 0.5-, 1-, 2-, and 3-degree buffers of the project study area, respectively (Figure 4). This study also considered three additional randomly sampled subsets of the 610 gages located within a three-degree buffer of the project study area of sizes 153, 305, and 458 precipitation gages, respectively (Figure 4). The total number of annual maxima observations for the 85, 151, 360, and 610 precipitation gages located within 0.5, 1, 2, and 3-degree buffers of the project study area, as shown in Figure 4a, were 5625, 10,347, 24,209, and 41,395, respectively. The total number of annual maxima observations for the 153, 305, and 458 precipitation gages shown in Figure 4b–d were 10,171, 20,829, and 31,154, respectively.

### 2.3. Gridded Covariate Data

Selected gridded covariate data included longitude (X), latitude (Y), elevation (Z), their products (XY, XZ, YZ), and climatological information extracted from the Parameter-elevation Relationships on Independent Slopes Model (PRISM) long-term mean monthly gridded data sets at a 30 arc-second resolution [54] for all gaged sites within each study region. MSP model deployments for the WRB study region applied the PRISM Norm81m long-term (1981–2010) mean monthly gridded data sets (seasonal precipitation, maximum/minimum/mean temperature, mean dew point temperature, minimum/maximum vapor pressure deficit); whereas the MSP deployments for the TRB study area applied the PRISM Norm91m long-term (1991–2020) mean monthly gridded data sets (annual precipitation, maximum/minimum/mean temperature, mean dew point temperature, minimum/maximum vapor pressure deficit, global shortwave solar radiation received on a horizontal/sloped surface, global shortwave solar radiation received on a horizontal surface under clear sky conditions, cloudiness). These covariates and their squares constituted the entire set of covariables (26/34 in total for the WRB/TRB study areas) considered to build trend surfaces associated with each MSP deployment for each study region. The selection of these covariates was based on past literature that demonstrated links between local extreme precipitation, physical information (e.g., elevation, climatology) [55–57], and rainfall-temperature thermodynamic relationships [58–60].

### 2.4. Methods

The following is a summary of the approach that was employed to develop and apply each distinct MSP model for the WRB and TRB study areas.

MSP model fitting was a two-step procedure involving trend surface and simple MSP model fitting and selection wherein each separate step assumed independence among the extreme data and unit Fréchet margins, respectively. The results from these two separate steps were subsequently combined to fit the full MSP model, wherein the trend surface and dependence parameters were fit simultaneously. The uncertainty of the MSP model was quantified through the use of the bootstrap method [35,36].

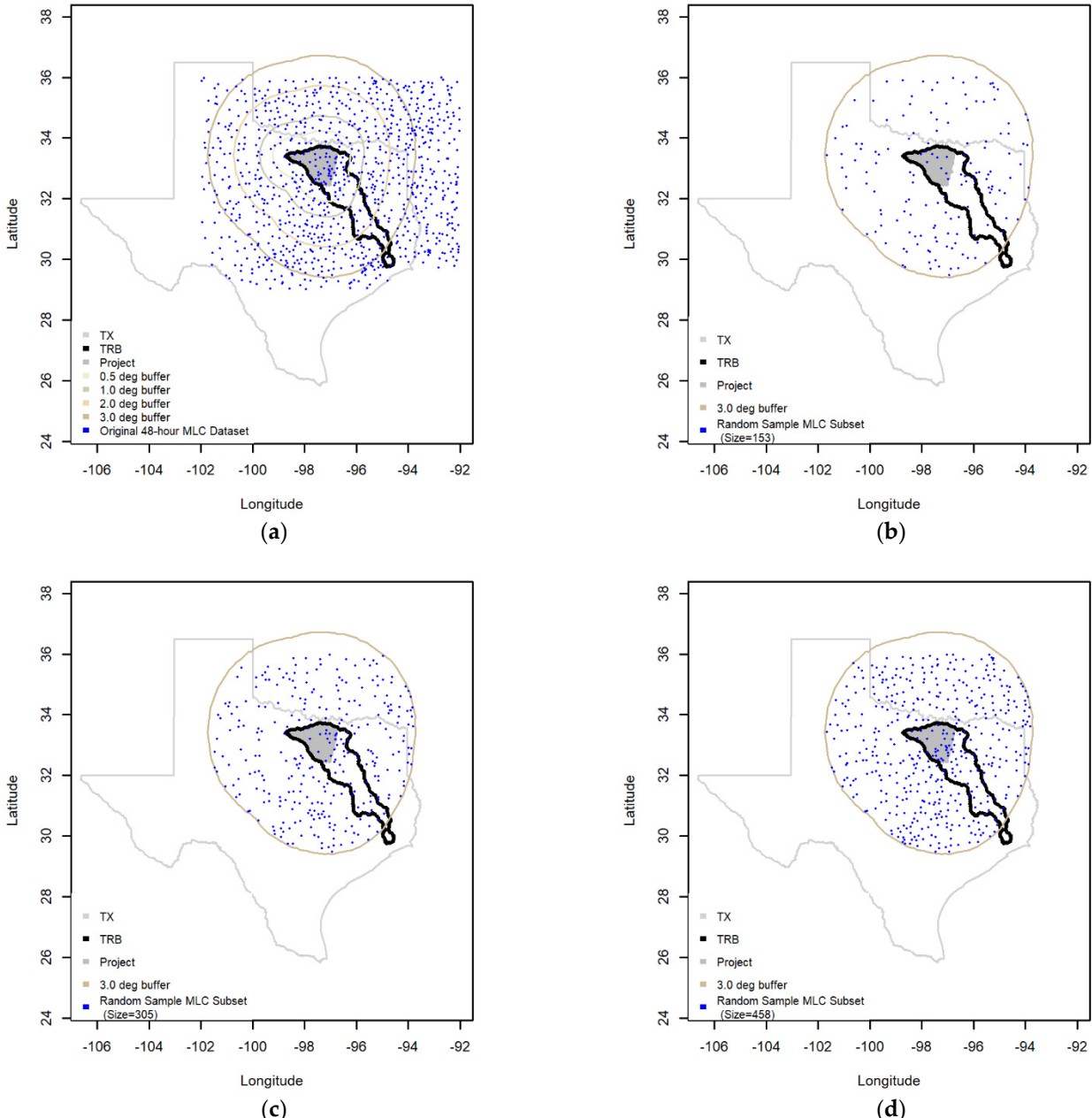

**Figure 4.** Precipitation gage subsets of the 48 h duration mid-latitude cyclone (MLC) annual maximum (July to June) precipitation dataset that was compiled for the eight-degree-by-six-degree rectangular region with north, south, east, and west extents of approximately 29° N, 35° N, −101° W, and −93° W [52]. (**a**) Four subsets of the original 48 h duration MLC annual maximum precipitation dataset included the 85, 151, 360, and 610 gages located within 0.5, 1, 2, and 3-degree buffers of the project study area, respectively. Randomly sampled subsets of the 610 gages located within a 3-degree buffer of the project study area of sizes (**b**) 153, (**c**) 305, and (**d**) 458. These precipitation gage datasets were prepared and used to develop and apply distinct MSPs to model 48 h duration MLC annual maximum in the Trinity River Basin (TRB) of Texas (TX). For each plot, the horizontal axis is in degrees longitude and the vertical axis is in degrees latitude.

The spectral representation of an MSP, introduced by de Haan [61] has resulted in the subsequent development of several parametric models for spatial extremes, with different distributional assumptions yielding different MSP models. Marginal distributions of an MSP model can be shown to be GEV distributed, with possibly different parameters by



location [13,14]. A simple MSP is defined to have unit Fréchet, rather than spatially variable GEV, margins.

For each MSP model employed, the extremal-t [62] simple MSP was utilized. However, in each case, five different correlation functions were considered, including the Bessel, Cauchy, generalized Cauchy, powered exponential, and Whittle–Matern correlation functions [63]. Model selection from among the five potential extremal-t MSP models was based on the composite likelihood information criterion, an adaption of the Takeuchi Information Criterion (TIC) [64], due to the application of the composite pairwise likelihood-based estimation approach [13,14]. The Schlather process was not considered since it is a special case of the extremal-t process [13,14]. The Smith process [65] was also not considered as its realizations are known to be too smooth for most practical applications [13,14]. The Brown–Resnick process [66,67] was also not considered since it is known to be difficult to work with [14]. To this end, Nicolet et al. [68] reported better performance for the extremal-t process relative to other available MSP models in their study of the dependence structure of extreme snowfall in the French Alps.

The extremal coefficient function is a convenient summary measure of dependence among extreme data [69]. In the bivariate case, assuming isotropy, the extremal coefficient is a function of the Euclidean distance, $h$, between any two spatial locations. It takes on values between one when the observations are fully dependent, and two, when they are independent [11,13,18]. Interestingly, there exists a unique mapping between the extremal coefficient function and the F-madogram [70], another well-defined measure of dependence among extreme data. Moreover, there exist empirical estimators for the F-madogram [10,14]. Inspection of plots of the extremal coefficient function for a model and its data is a recommended qualitative evaluation of a fitted MSP model [13,33].

As previously mentioned, the marginal distributions of an MSP are GEV distributed, possibly varying by location. The composite (pairwise) likelihood-based approach often used to fit a simple MSP model can be readily adapted to accommodate the simultaneous estimation of trend surface and dependence parameters [13,14,33]. Trend surfaces are functions that use spatially varying covariates to model the location, $\mu(x)$, scale, $\sigma(x)$, and shape, $\xi(x)$, parameters of the GEV marginal distributions. For example, linear trend surfaces are of the form $\mu(x) = \eta_{\mu,0} + \eta_{\mu,1}cov_{\mu,1} + \ldots + \eta_{\mu,n_\mu}cov_{\mu,n_\mu}$, $\sigma(x) = \eta_{\sigma,0} + \eta_{\sigma,1}cov_{\sigma,1} + \ldots + \eta_{\sigma,n_{\mu\sigma}}cov_{\sigma,n_\sigma}$, $\xi(x) = \eta_{\xi,0} + \eta_{\xi,1}cov_{\xi,1} + \ldots + \eta_{\xi,n_\xi}cov_{\xi,n_\xi}$, where $\eta_{\cdot,i}$ and $cov_{\cdot,i}$ are the parameters and covariates of the linear trend surface for $\mu(x)$, $\sigma(x)$, and $\xi(x)$, respectively. Potential covariates include; for example, gridded physiographic (e.g., such as x-location, y-location, elevation, slope, aspect, curvature) and climatological (e.g., such as mean annual/monthly temperature, precipitation, wind, solar radiation) data. Cooley et al. [10] described trend surface modeling as capturing regional spatial effects (i.e., climate effects) and that local spatial effects (i.e., weather effects) are best described by a stochastic dependence structure.

A novel approach was applied in this study to develop linear trend surfaces for the marginal parameters [37]. Zou and Hastie [71] introduced the elastic-net penalty as a compromise between ridge [72,73] and lasso [74] regression. Given observations $y_i, i = 1, \ldots, n$, an $n \times m$ matrix of normalized covariates $X$, and an assumed linear model $y_i = \eta_0 + \eta_1 x_{i,1} + \ldots + \eta_m x_{i,m}$, the elastic net minimizes $\frac{1}{2n}\sum_{i=1}^{n} \widetilde{w}_i(y_i - \eta_0 - \eta x_i^T)^2 + \lambda\sum_{j=1}^{m}\left[\frac{1}{2}(1-\alpha)\eta_j^2 + \alpha|\eta_j|\right]$, where $\lambda$ is a non-negative regularization parameter that is tuned to weigh the overall strength of the penalty and $\alpha \in [0,1]$ is specified to control the penalty term to vary from ridge regression at $\alpha = 0$ to lasso regression at $\alpha = 1$, and $\widetilde{w}_i$ is the weight assigned to the ith observation [41]. Ridge regression yields smooth solutions that include all the predictors; whereas, the application of lasso regression results in automatic variable selection; i.e., sparse, much more easily interpretable solutions [75]. The elastic net mixes the two methods. As $\alpha$ increases from 0 to 1 for a fixed $\lambda$, the number of zero-valued $\eta_j$ increases from 0 to the sparsity of the lasso [41].

In this study, variable selection was preferred and $\alpha$ was specified close to 1 ($\alpha = 0.95$) for numerical stability [41]. The $\widetilde{w}_i$ were normalized and assigned in proportion to the number of observations at each site. Cross-validation (CV) was applied to ensure that the minimizing value for $\lambda$ was properly located for each elastic net application. A set of base covariates and their squares [18] constituted the entire set of covariables considered to build each trend surface. Davison and Gholamrezaee [18] demonstrated significant improvements in model fits with the inclusion of quadratic terms for modeling the marginal distributions in their MSP-based spatial analysis of extreme temperature data in Switzerland. Independent elastic net application results for $\mu(x)$ and $\sigma(x)$, with $\xi(x) = \xi$, guided subsequent spatial GEV model formulation and evaluation. In spatial modeling studies of this type, it is not uncommon to treat the GEV shape parameter in this manner [18,76]. The log-likelihood of the spatial GEV model, which assumes independence among the sample observation sites, is given by $l\big(\eta_\mu, \eta_\sigma, \eta_\xi\big) = \sum_{i=1}^{nsite} \sum_{j=1}^{nobs} \Big\{ -log\sigma_i - \Big( 1 + \xi_i \frac{y_{i,j} - \mu_i}{\sigma_i} \Big)^{-1/\xi_i} - \Big( 1 + \frac{1}{\xi_i} \Big) log \Big( 1 + \xi_i \frac{y_{i,j} - \mu_i}{\sigma_i} \Big) \Big\}$, where $\mu_i$, $\sigma_i$, and $\xi_i$ are the GEV location, scale, and shape parameters for the $i$-th site and $y_{i,j}$ is the $j$-th observation for the $i$-th site [37].

The joint spatial modeling of precipitation observations enables the computation, via Monte Carlo simulation, of an integral such as $I = \frac{1}{|\mathcal{B}|} \int_{\mathcal{B}} Z(x) dx$ for a basin $\mathcal{B}$ to estimate the unknown distribution of the random variable $I$ (e.g., areal average precipitation exceedance estimate) [33]. Areal exceedances were calculated by simulating multiple independent copies of the MSP for an area of interest [12,13,33], using its fitted values or random samples either gleaned from model calibration [10,13,18,26,34] or application of the bootstrap method [35,36]. Schlather [77] introduced an approach for simulating an independent realization of a simple MSP with only a finite number of replicates.

Distinct MSP models were developed and applied to examine the impacts of study area sampling density, precipitation gage extent and density, and model fitting method on areal exceedance calculations for a project area within the WRB (Figure 1) and TRB (Figure 2), respectively. Each analysis involved a comparison of calculated areal exceedance values for AEPs of $10^{-1}$, $10^{-2}$, $10^{-3}$, and $10^{-4}$. Seven distinct models were developed for both the WRB and the TRB. Salient details for these models are summarized in Table 1. Two different fitting methods were considered for the WRB models whereas all seven models deployed for the TRB used the same approach for model optimization. Except for WMSP04, all models were fit using observed data. The model WMSP04 was fit using synthetic data that was generated from the spatial process WMSP02.

**Table 1.** Summary details for each MSP that was developed and applied to calculate areal exceedances for a project area within the Willamette River Basin (WRB) and the Trinity River Basin (TRB), respectively.

| Model Name | Dataset | MSP Model Description |
|---|---|---|
| | | WRB MSP Models |
| WMSP01 | Figure 3a | General MSP was fit using constrained optimization and initialized with simple MSP and spatial GEV trend surface parameter estimates |
| WMSP02 | Figure 3a | General MSP was fit using unconstrained optimization and initialized with simple MSP and spatial GEV trend surface parameter estimates |
| WMSP03 | Figure 3b | General MSP was fit using unconstrained optimization and initialized with simple MSP and spatial GEV trend surface parameter estimates |
| WMSP04 | Figure 3c | |
| WMSP05 | Figure 3d | General MSP was fit using constrained optimization and initialized with simple MSP and spatial GEV trend surface parameter estimates |
| WMSP06 | Figure 3e | General MSP was fit using constrained optimization and initialized with simple MSP and spatial GEV trend surface parameter estimates |
| WMSP07 | Figure 3f | |

**Table 1.** *Cont.*

| Model Name | | MSP Model Description |
|---|---|---|
| | **Dataset** | |
| | | TRB MSP Models |
| TMSP01 | Figure 4a | Used the 85 precipitation gages within the 0.5° buffer of the project area |
| TMSP02 | Figure 4a | Used the 151 precipitation gages within the 1° buffer of the project area |
| TMSP03 | Figure 4a | Used the 360 precipitation gages within the 2° buffer of the project area |
| TMSP04 | Figure 4a | Used the 610 precipitation gages within the 3° buffer of the project area |
| TMSP05 | Figure 4b | A random sample of 153 (~25%) of the 610 precipitation gages within the 3° buffer of the project area |
| TMSP06 | Figure 4c | A random sample of 305 (50%) of the 610 precipitation gages within the 3° buffer of the project area |
| TMSP07 | Figure 4d | A random sample of 458 (~75%) of the 610 precipitation gages within the 3° buffer of the project area |

2.4.1. Impact of MSP Areal Exceedance Calculations to Study Area Sampling Density

The fitted values for the max-stable model WMSP01 were used to simulate independent copies of the spatial process for the four study area sampling densities of the 536 square kilometers Cougar Dam project study area shown in Figure 5. Figure 5a depicts the delineated area at a 30 arc-second grid cell scale, the grid scale of the covariate data that were used for the trend surface modeling analyses. It consisted of 853 grid cells that were equally weighted for areal exceedance calculations. Figure 5b–d depict the subsets of 213 (approximately 25% of the 853), 85 (approximately 10% of the 853), and 21 (approximately 2.5% of the 853) points that were randomly sampled from the original 853 grid cell points shown in Figure 5a, and their corresponding Voronoi polygons which were computed using the public domain Geographic Information System QGIS [78]. For these study area sampling densities, each sampled point's simulated value was weighted using its corresponding Voronoi cell area. One hundred thousand values of the fitted process WMSP01 were simulated at each of the 853 grid cell points shown in Figure 5a to compute areal exceedances for the Cougar Dam project study area. One million values were simulated for the three coarsened study area sampling densities shown in Figure 5b–d. The 1,000,000 values were obtained from 10 separate runs that each simulated 100,000 values.

The subsets of 213, 85, and 21 points that were sampled from the 853 grid cell points that compose the entire project study area were random and arbitrary (Figure 5). A random sample subset optimized for areal exceedance calculation was hypothesized to be one whose distribution of pointwise return level values best matched its comparable distribution for the entire set of 853 grid cells that compose the study area. Directed subsets of sizes 213, 85, and 21 were iteratively computed to find an optimized random sample in each case. Ten thousand samples were evaluated for each of the three study area sampling densities. For an AEP of $10^{-3}$, the distribution of pointwise return level values was defined to be the vector of quantile values for the predetermined probability values of 0.01, 0.1, 0.2, 0.3, 0.4, 0.5, 0.6, 0.7, 0.8, 0.9, and 0.99. For each study area sampling density, the optimized subset was the one among the 10,000 evaluated with the minimum sum of squared differences. This approach to reducing the study area sampling density for areal exceedance calculation differed from working with a coarsened grid cell scale which is known to dampen extremes [79–81].

A delineation of the 15,662 square kilometer TRB project study area composed of the Upper West Fork Trinity (12030101; 5102 square kilometers), Lower West Fork Trinity (12030102; 3911 square kilometers), Elm Fork Trinity (12030103; 4766 square kilometers), and Denton (12030104; 1883 square kilometers) USGS 8-digit HUC sub-basins (Figure 2) consisted of 21,794 grid cells at a 30 arc-second grid cell scale. The fitted values for the

max-stable model TMSP04 were used to simulate five independent copies of the spatial process for the entire study area and the two study area sampling densities shown in Figure 6. Figure 6a,b depict the subsets of 2180 (approximately 10% of the 21,794) and 218 (approximately 1% of the 21,794) points that were randomly sampled from the original 21,794 grid cell points, and their corresponding Voronoi polygons. As was performed for the Cougar dam project area in the WRB, for these TRB study area sampling densities, each sampled point's simulated value was weighted using its corresponding Voronoi cell area. For each of the five simulated copies of the TMSP04 spatial process, areal means were computed for the three project study area spatial resolutions.

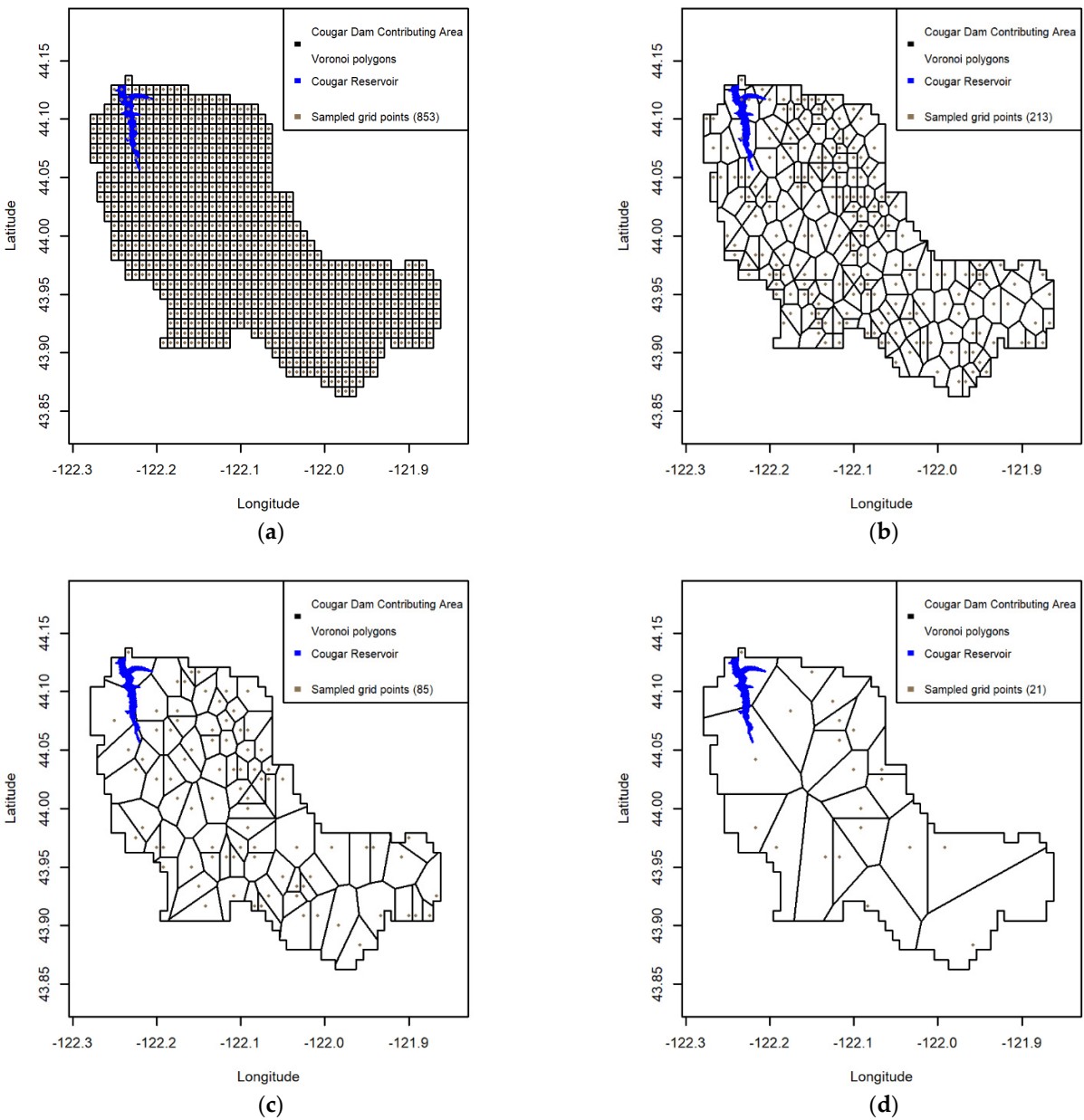

**Figure 5.** (**a**) A delineation of the Willamette River Basin's 536 square kilometer Cougar dam project study area consisted of 853 grid cell points at the 30 arc-second scales. Subsets of (**b**) 213 (approximately 25% of the 853), (**c**) 85 (approximately 10% of the 853), and (**d**) 21 (approximately 2.5% of the 853) points that were randomly sampled from the original 853 grid cell points shown in (**a**), and their corresponding Voronoi polygons. For each plot, the horizontal axis is in degrees longitude and the vertical axis is in degrees latitude.

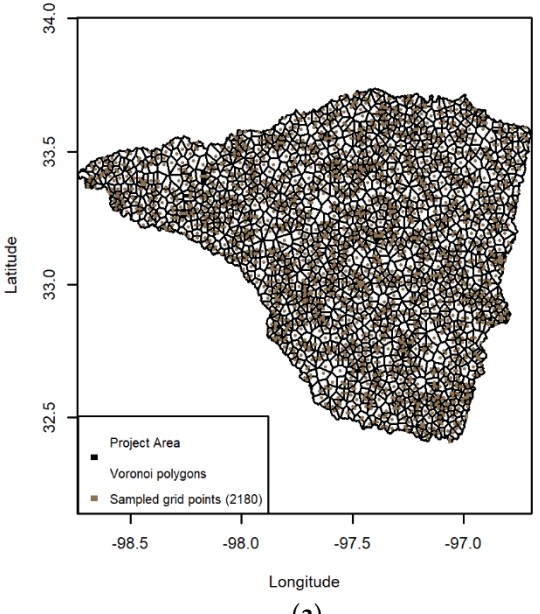
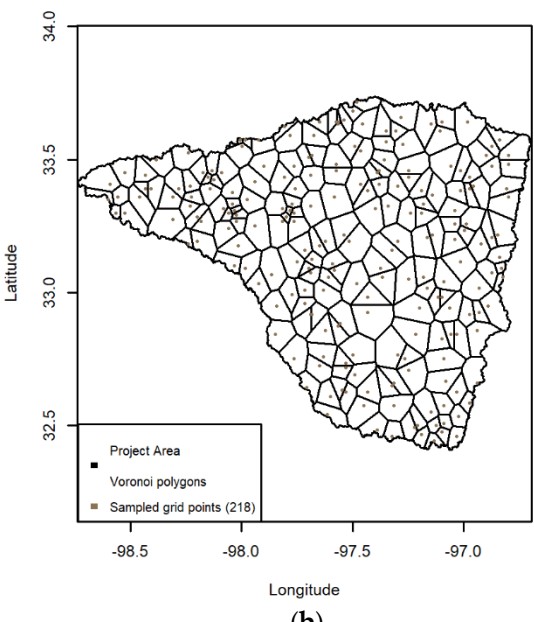

**Figure 6.** For the Trinity River Basin project area, subsets of (**a**) 2180 (approximately 10% of the 21,794) and (**b**) 218 (approximately 1% of the 21,794) points were randomly sampled from the 21,794 grid cell points that composed the project study area's delineation at the 30 arc-second grid cell scale, and their corresponding Voronoi polygons. For each plot, the horizontal axis is in degrees longitude and the vertical axis is in degrees latitude.

### 2.4.2. Impact of MSP Areal Exceedance Calculations to Precipitation Gage Extent

The fitted values for the max-stable models WMSP02, WMSP03, and WMSP04 were used to simulate one million independent copies for each spatial process for the 536 square kilometer Cougar Dam project study area using the sampling density shown in Figure 5c.

The max-stable model WMSP04 was fit using synthetic data that was generated from the spatial process WMSP02. The fitted parameter estimates for WMSP02 were applied to simulate 152 independent storms for the 1-degree by 1-degree box located within the WRB with north, south, east, and west extents of 44.75° N, 43.75° N, −121.8° W, and −122.8° W, respectively. One of the 152 independent storms simulated for this box region is depicted in Figure 7. Twenty-six of the one-hundred and forty precipitation gages shown in Figure 3a were located within this box region (Figure 3c). For each of the 152 storms, observations were extracted at each of the 26 precipitation gage sites shown in Figure 3c. For each of the 26 precipitation gage sites shown in Figure 3c, whenever its original precipitation dataset had a missing value, a missing data value designation replaced that year's synthetic storm observation.

The fitted values for the max-stable models TMSP01, TMSP02, TMSP03, and TMSP04 were used to simulate one million independent copies for each spatial process for the 15,662 square kilometer TRB project study area using the sampling density shown in Figure 6b.

### 2.4.3. Impact of MSP Areal Exceedance Calculations to Precipitation Gage Density

The fitted values for the max-stable models WMSP01, WMSP05, WMSP06, and WMSP07 were used to simulate one million independent copies for each spatial process for the 536 square kilometer Cougar Dam project study area using the sampling density shown in Figure 5c.

The fitted values for the max-stable models TMSP04, TMSP05, TMSP06, and TMSP07 were used to simulate one million independent copies for each spatial process for the 15,662 square kilometer TRB project study area using the sampling density shown in Figure 6b.



### 2.4.4. Impact of MSP Areal Exceedance Calculations on Model Fitting Method

The MSPs WMSP01 and WMSP02 were each fit using different optimization methods. While the optimization for each model was initialized using its corresponding simple MSP and spatial GEV trend surface parameter estimates, WMSP01 was calibrated using constrained optimization whereas unconstrained optimization was used to fit WMSP02. The fitted values for each max-stable model were used to simulate ten million independent copies of each spatial process for the 536 square kilometer Cougar Dam project study area using the sampling density shown in Figure 5c.

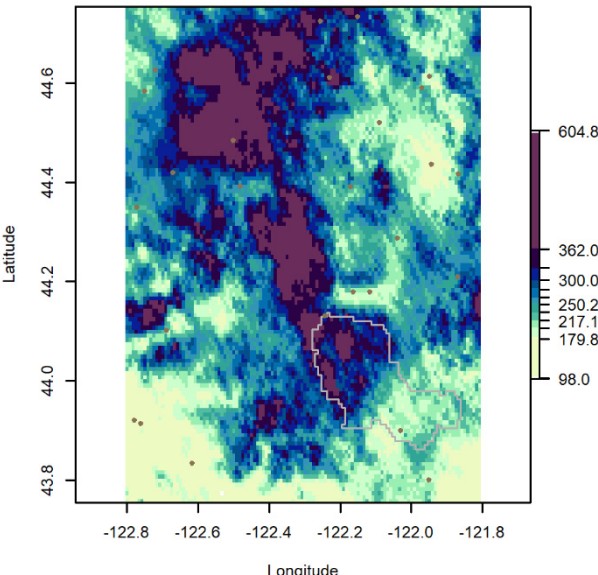

**Figure 7.** A cool season (October to April) three-day duration storm simulated for the one-degree-by-one-degree box located within the Willamette River Basin with north, south, east, and west extents of 44.75° N, 43.75° N, −121.8° W, and −122.8° W, respectively. Simulated precipitation values are in mm. The storm was simulated using fitted model parameter estimates for WMSP02 (Table 1), an extremal-t MSP with Whittle–Matern correlation function. Twenty-six of the 140 original precipitation gage sites that exist within the box region (Figure 3a,c), including the Cougar Dam project study area, are also shown. The horizontal axis is in degrees longitude and the vertical axis is in degrees latitude.

## 3. Results and Discussion

Aside from Coles and Tawn [20] and Davison et al. [11], little attention has been given to practical areal precipitation-frequency estimation using MSPs [19–32,82]. Rather than areal exceedance calculations, practice directed MSP applications have focused on conditional simulations of MSPs [82], covariate selection for estimation of the GEV marginal distributions [21], pointwise return level estimation [22,32], Intensity-Duration-Frequency curve development [23], conditional maps of pointwise return levels across different durations [24], general overviews of the MSP modeling process [11,25,26], inference method development for fitting MSPs [27,28,31], and the use of a climate index for trend surface development [29,30]. Azizah et al. [19] studied the application of the Broyden-Fletcher Goldfarb-Shanno (BFGS) Quasi-Newton method for general MSP parameter estimation while considering the Smith model [65]. Coles and Tawn [20] derived a closed-form GEV distribution solution to calculate areal exceedances that combined a GEV marginal trend surface, wherein only the GEV location and shape parameters were allowed to vary spatially, with an MSP model of spatial dependence. However, their areal exceedance calculations were dependent upon a computation-intensive areal coefficient that characterized the extremal spatial dependence but was not completely independent of marginal behavior. Davison et al. [11] demonstrated, based on a limited set of simulations, applications of the Smith [65], Schlather [13,14], extremal-t [13,14], and Brown–Resnick [66,67] MSPs for

areal precipitation frequency estimation. Neves and Gomes [25] and Padoan et al. [27] both mentioned that guidance for modeling spatial extremes was limited.

### 3.1. Impact of MSP Areal Exceedance Calculations to Study Area Sampling Density

Table 2 lists areal exceedance values that were calculated using the fitted values from the max-stable model WMSP01 for the three study area sampling densities of the 536 square kilometer Cougar Dam project study area shown in Figures 5 and 8.

**Table 2.** Areal exceedance values, in inches, were calculated using the fitted values from the max-stable model WMSP01 for the 536 square kilometer Cougar Dam project study area shown in Figures 5 and 8 using four study area sampling densities. Row 1 results were based on 100,000 independent realizations of WMSP01. The results in rows 3/4, 6/7, and 9/10 list the average, minimum, and maximum values obtained from ten distinct runs that each computed 100,000 independent realizations of WMSP01. The results in rows 2, 5, and 8 were obtained from the 1,000,000 independent realizations of WMSP01. Row 3, row 6, and row 9 results were obtained using arbitrary random sampling whereas the results in row 4, row 7, and row 10 used optimized sample subsets.

| Sampling Density | | AEP | | | |
|---|---|---|---|---|---|
| | | $10^{-1}$ | $10^{-2}$ | $10^{-3}$ | $10^{-4}$ |
| 100% (853 points) | 1 | 9.71 | 13.47 | 17.50 | 21.55 |
| 25% (213 points) | 2 | 9.74 | 13.44 | 17.22 | 21.23 |
| | 3 | 9.74 (9.72, 9.78) | 13.44 (13.35, 13.52) | 17.21 (16.95, 17.52) | 21.27 (20.01, 21.81) |
| | 4 | 9.73 (9.72, 9.76) | 13.44 (13.34, 13.51) | 17.22 (16.95, 17.55) | 21.20 (20.06, 21.96) |
| 10% (85 points) | 5 | 9.69 | 13.41 | 17.27 | 21.27 |
| | 6 | 9.69 (9.67, 9.72) | 13.41 (13.34, 13.53) | 17.27 (17.03, 17.62) | 21.34 (20.42, 22.08) |
| | 7 | 9.68 (9.65, 9.70) | 13.39 (13.33, 13.50) | 17.22 (17.03, 17.51) | 21.16 (20.20, 21.75) |
| 2.5% (21 points) | 8 | 9.77 | 13.49 | 17.30 | 21.45 |
| | 9 | 9.77 (9.75, 9.80) | 13.50 (13.43, 13.60) | 17.30 (17.09, 17.55) | 21.49 (20.82, 22.61) |
| | 10 | 9.74 (9.71, 9.76) | 13.45 (13.39, 13.54) | 17.26 (16.98, 17.59) | 21.48 (20.55, 22.60) |

Figure 8 shows the $10^{-3}$ AEP pointwise return level values that were computed for the entire study area at the 30 arc-second grid cell scale using the fitted values from the max-stable model WMSP01. In Figure 8, the optimized random subsets that were computed for the study area sampling densities of sizes 213, 85, and 21, respectively, overlay the pointwise return level map computed for the study area.

Table 3 lists quantile values that were extracted from the $10^{-3}$ AEP pointwise return level grid of the Cougar Dam study area for the four sampling densities. It also reports the sum of the squared difference (SSD) value that was computed for each random subset. The SSD value computed for each random subset compared quantile values from the full set of 853 grid points composing the study area. For the subset that used 213 (approximately 25%) of the entire set of 853 grid points composing the study area, Table 3 lists the quantile values that were extracted from the arbitrary random sample (row 2; Figure 5b) and the optimized random sample (row 3; Figure 8a). It also lists those values, in row 4/6 and row 5/7, for the subset that used 85/21 (approximately 10%/2.5%) of the study area's 853 grid points. Row 4 and row 6 report the extracted vector of quantile values, and its computed SSD, for the arbitrary random sample whereas row 5 and row 7 list corresponding values

for the directed random sample. As measured by the SSD values reported in Table 3, the optimized random subsets better characterized, throughout the study area, the modeled $10^{-3}$ AEP pointwise return levels than their corresponding arbitrary random samples. Among the 10,000 random samples of size 213 that were evaluated to find an optimized subset, the minimum, first quartile, median, third quartile, and maximum computed SSD values were 0.015, 0.180, 0.320, 0.571, and 3.77, respectively. For the 10,000 random samples of size 85/21, those values were 0.028/0.156, 0.565/2.69, 1.01/4.44, 1.71/7.56, and 13.51/49.89, respectively.

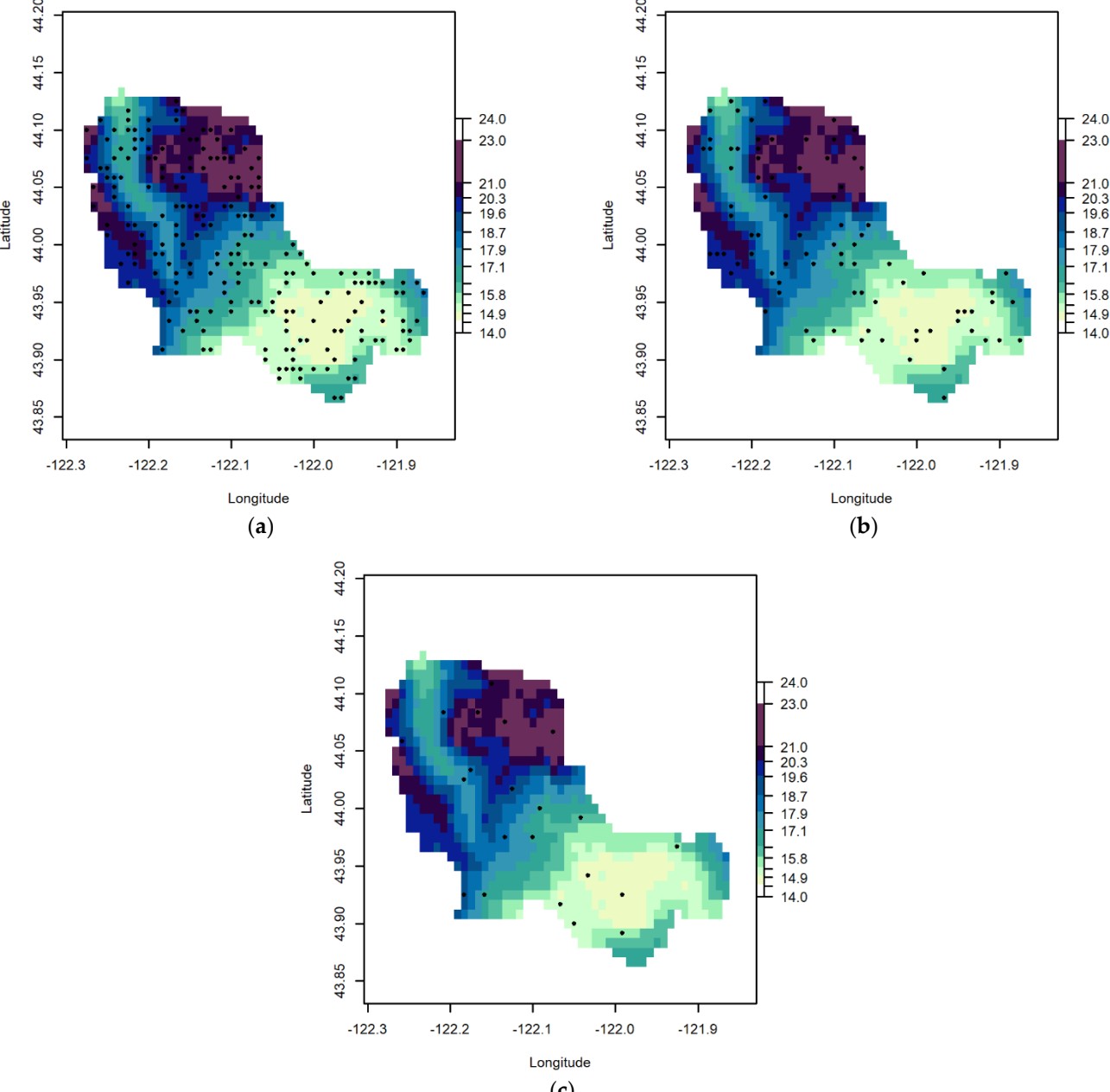

**Figure 8.** The $10^{-3}$ AEP pointwise return level values, in inches, were computed for the Cougar dam study area at the 30 arc-second grid cell scale using the fitted values from the max-stable model WMSP01. The optimized random subsets that were computed for the study area sampling densities of sizes (**a**) 213, (**b**) 85, and (**c**) 21, respectively, overlay the pointwise return level map computed for the study area. For each plot, the horizontal axis is in degrees longitude and the vertical axis is in degrees latitude.

**Table 3.** Quantile values, in inches, were extracted from the $10^{-3}$ AEP pointwise return level grid of the Cougar dam study area for the four sampling densities. Row 2, row 4, and row 6 results were obtained using arbitrary random sampling whereas the results in row 3, row 5, and row 7 used optimized sample subsets (SSD = sum of squared difference).

| Study Area Sampling Density | | Probability | | | | | | | | | | | |
|---|---|---|---|---|---|---|---|---|---|---|---|---|---|
| | | 0.01 | 0.1 | 0.2 | 0.3 | 0.4 | 0.5 | 0.6 | 0.7 | 0.8 | 0.9 | 0.99 | SSD |
| 100% | 1 | 14.70 | 14.94 | 15.43 | 15.97 | 16.64 | 17.50 | 18.38 | 19.34 | 20.19 | 20.96 | 22.35 | |
| 25% | 2 | 14.71 | 14.98 | 15.46 | 16.18 | 16.99 | 17.96 | 18.99 | 19.67 | 20.41 | 21.07 | 22.41 | 0.94 |
| | 3 | 14.75 | 14.96 | 15.45 | 15.96 | 16.64 | 17.44 | 18.44 | 19.28 | 20.20 | 20.99 | 22.34 | 0.015 |
| 10% | 4 | 14.74 | 14.98 | 15.46 | 16.54 | 17.38 | 18.18 | 19.14 | 19.72 | 20.62 | 20.98 | 22.23 | 2.26 |
| | 5 | 14.70 | 15.01 | 15.45 | 16.01 | 16.69 | 17.52 | 18.47 | 19.35 | 20.16 | 21.00 | 22.26 | 0.028 |
| 2.5% | 6 | 15.17 | 15.45 | 16.20 | 17.01 | 17.63 | 19.04 | 19.64 | 20.22 | 21.01 | 21.53 | 22.13 | 8.91 |
| | 7 | 14.75 | 14.95 | 15.46 | 16.19 | 16.89 | 17.63 | 18.42 | 19.30 | 20.11 | 21.05 | 22.42 | 0.156 |

The first row of Table 2 lists areal exceedance estimates obtained from the 100,000 simulations that were performed using the entire set of 853 grid cells that composed the Cougar dam study area (Figure 5a). The following three rows (2–4) list the estimates obtained from the sampling density that is considered an arbitrary (Figure 5b) and optimized (Figure 8a) random subset of size 213 from the 853 grid cells that composed the study area. Row 2 lists the areal exceedances that were calculated from the 1,000,000 simulations that used the arbitrary random sample (Figure 5b). For a direct comparison with the areal values reported in row 1, for each AEP, row 3 lists the average areal exceedance value obtained from the ten runs that each involved one hundred thousand simulations while using the arbitrary random subset. The values reported in row 4 are like those listed in row 3; however, they were obtained using the optimized random subset. Rows 3 and 4 also include summaries of the minimum and maximum reported areal exceedance values across the 10 simulations. The areal exceedance values reported in rows 5–7/8–10 are like those reported in rows 2–4; however, they were obtained using an arbitrary (Figure 5c/Figure 5d) and optimized (Figure 8b/Figure 8c) random subset of size 85/21 from the 853 grid cells that composed the study area.

The areal exceedance values calculated for the study area sample subsets agreed well with their comparable values obtained using the entire set of 853 grid cell points that composed the Cougar Dam contributing drainage area (Table 2). At the $10^{-1}$, $10^{-2}$, $10^{-3}$, and $10^{-4}$ AEP, the average values reported for the arbitrarily sampled subsets of 213/85/21 points differed from the values reported for the full grid of 853 points by 0.31/0.21/0.62, 0.22/0.45/0.22, 1.67/1.32/1.15, and 1.31/0.98/0.28 percent, respectively. For the optimized sample subsets, these values were 0.21/0.31/0.31, 0.22/0.60/0.15, 1.61/1.61/1.38, and 1.64/1.83/0.33 percent, respectively. Except for the $10^{-1}$ AEP, the values reported for the full grid of 853 points were also within the range of values reported for each of the two subsets, regardless of whether the sample subset was arbitrary or optimized.

The results summarized in Table 2 were highly encouraging from the practical perspective of the compute time required to obtain a usable mean areal exceedance curve, including uncertainty quantification. Using code and a compute resource (i.e., a laptop computer with a 12th Gen Intel(R) Core (TM) i9-12900HK processor and 64.0 GB installed RAM) in the same way, Figure 9 summarized the times required to compute different sized sets of independent realizations of the MSP WMSP01 for the entire grid of 853 points composing the Cougar dam study area and its random sample subsets of sizes 213, 85, and 21, respectively. From Figure 9, 287.84, 5.34, 0.59, and 0.04 min were required to compute one hundred thousand independent realizations of the MSP WMSP01 for the Cougar dam study area sampling densities that consisted of 853, 213, 85, and 21 grid cell points.

It required 5.22/5.59 min to compute 1/12 million independent realizations of the MSP WMSP01 using the study area sampling density that consisted of 85/21 grid cell points. A similar amount of time (5.53 min) was required to compute two thousand independent realizations of WMSP01 while using the 853 grid cell points that composed the Cougar Dam study area.

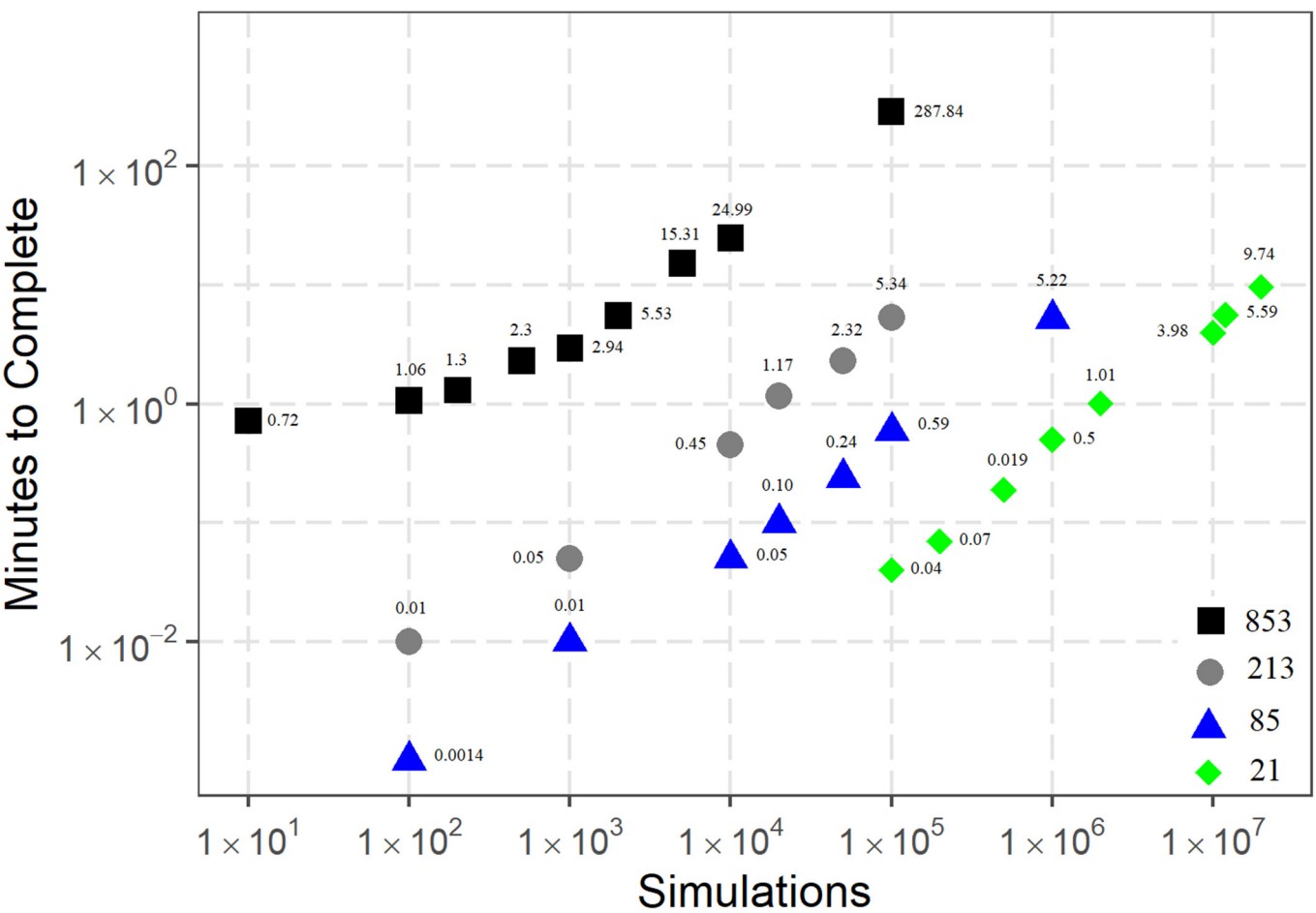

**Figure 9.** The times required to compute different-sized sets of independent realizations of the MSP WMSP01 for the entire grid of 853 (black square) points composing the Cougar Dam study area and its random sample subsets of sizes 213 (gray circle), 85 (blue triangle), and 21 (green rhombus), respectively.

Significant reductions in compute times were achieved by applying sample subsets rather than the entire grid when calculating areal exceedances with the MSP WMSP01 for the Cougar dam study area (Figure 9). These reductions were achieved with little loss in accuracy (Table 2). These results, albeit limited, challenge the assessment that a limitation for practical applications of MSP models is their intensive computational requirements [83].

The application of an optimized subset did not provide any benefit relative to the use of an arbitrary sample, at least for the sampling densities and AEPs that were evaluated for the Cougar Dam study area (Table 2). However, further study is encouraged, directed at alternative approaches for optimized subset selection, for study areas with different sizes and locations, and for a wider range of AEPs and sampling densities. A comparison of results from the study area sampling approach applied herein with one that coarsens the base grid cell scale to reduce the study area grid density is another potential path for future related studies.

Only five independent realizations of the MSP TMSP04 were computed given the size of the TRB project study area (15,662 square kilometers) and the number of grid cells

that compose it at the 30 arc-second scales (21,794). Areal means were computed for the three TRB study area sampling densities for each independent realization of TMSP04 (Table 4). For each individual realization, the agreement among the computed areal means across the 3 study area sampling densities was excellent. Figure 10 shows the $10^{-3}$ AEP pointwise return level values that were computed throughout the entire TRB study area at the 30 arc-second grid cell scale using the fitted values from the max-stable model TMSP04. One likely explanation for the excellent agreement among the areal means computed across the three TRB study area spatial densities is the relatively narrow range and predictable spatial pattern of the pointwise return level values throughout the TRB study area (Figure 10). For the TRB study area, more sampling density comparisons are needed, either more like those provided in Table 4 or better, like the ones provided in Table 2 for the Cougar dam study area in the WRB.

**Table 4.** For the three TRB study area sampling densities, the areal means, in inches, were computed for five independent realizations of TMSP04.

| | Areal Means | | | | |
|---|---|---|---|---|---|
| **Sampling Density** | **1** | **2** | **3** | **4** | **5** |
| 100% | 3.61 | 2.69 | 3.56 | 2.99 | 2.57 |
| 10% | 3.60 | 2.68 | 3.54 | 2.99 | 2.57 |
| 1% | 3.59 | 2.66 | 3.63 | 3.00 | 2.59 |

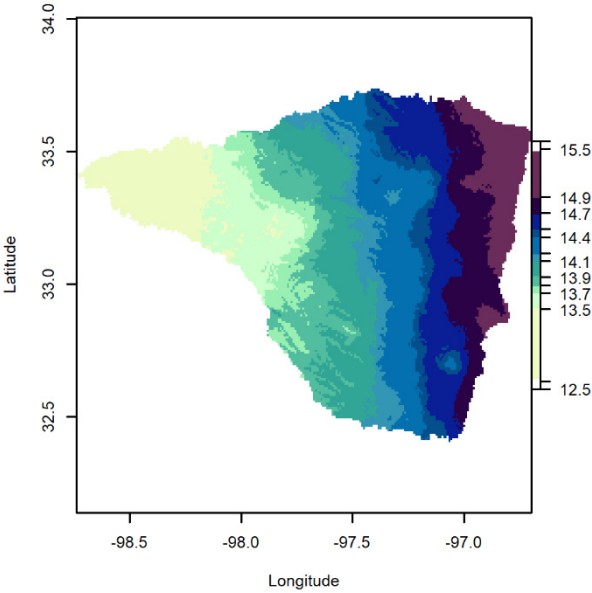

**Figure 10.** The $10^{-3}$ AEP pointwise return level values, in inches, were computed throughout the TRB study area at the 30 arc-second grid cell scale using the fitted values from the max-stable model TMSP04. The horizontal axis is in degrees longitude and the vertical axis is in degrees latitude.

### 3.2. Impact of MSP Areal Exceedance Calculations to Precipitation Gage Extent

Application of the three max-stable models that were developed for the WRB with three different gage extents resulted in relatively similar areal exceedance estimates for the Cougar dam study area in the WRB (Table 5). The model WMSP02 was developed using the data from the 140 gages from Skahill et al. [49] with at least 20 seasonal maxima observations [50] that were located within the WRB and a 20-km buffer of the WRB watershed boundary (Figure 3a). The model WMSP03 was developed using the data from the 286 precipitation gages from Skahill et al. [49] with at least 20 seasonal maxima observations whose footprint intersected with the precipitation gages that were used for a relatively recent L-moments RFA that was performed for the WRB [51] (Figure 3b). The model WMSP04

used the 26 precipitation gages located within the one-degree by-one-degree box with north, south, east, and west extents of 44.75° N, 43.75° N, −121.8° W, and −122.8° W, respectively (Figure 3c). The model WMSP04 was fit using synthetic data that was generated from the spatial process WMSP02. For WMSP04 and each of its 26 precipitation gage sites shown in Figure 3c, whenever the precipitation dataset for WMSP02 had a missing value, a missing data value designation replaced that year's synthetic storm observation. The areal exceedance estimates listed in Table 5 were obtained from one million independent copies for each spatial process, using their respective fitted values, for the 536 square erkilometer Cougar Dam project study area using the sampling density shown in Figure 5c.

**Table 5.** Areal exceedance values, in inches, calculated using the fitted values from the max-stable models WMSP02, WMSP03, and WMSP04 for the 536 square kilometer Cougar Dam project study area shown in Figures 5 and 8 using three gage extents (Figure 3a–c). The areal exceedance estimates were obtained from one million independent copies for each spatial process using the sampling density shown in Figure 5c.

| | AEP | | | |
|:---:|:---:|:---:|:---:|:---:|
| **Gage Extent (Model)** | $10^{-1}$ | $10^{-2}$ | $10^{-3}$ | $10^{-4}$ |
| WMSP02 | 9.66 | 13.49 | 17.55 | 22.03 |
| WMSP03 | 9.54 | 13.48 | 17.83 | 22.42 |
| WMSP04 | 9.53 | 13.33 | 17.4 | 21.54 |

Four MSP models were developed for the TRB with four different gage extents. The four models were developed using the annual maxima series from the 85, 151, 360, and 610 precipitation gages located within 0.5, 1, 2, and 3 degrees of the project study area, respectively (Figure 4a). The areal exceedance estimates calculated for the TRB project study area using these four MSP models were summarized in Table 6. The areal exceedance estimates listed in Table 6 were obtained from one million independent copies for each spatial process, using their respective fitted values, for the 15,662 square kilometer TRB project study area using the sampling density shown in Figure 6b. Across these four MSP models, there was greater variation among the areal exceedance estimates for the TRB study area than the areal exceedances reported in Table 5 for the Cougar dam safety project study area in the WRB, particularly for the AEP values of $10^{-3}$ and $10^{-4}$.

**Table 6.** The areal exceedance estimates, in inches, were obtained from one million independent copies for each spatial process, using their respective fitted values, for the 15,662 square kilometer TRB project study area using the sampling density shown in Figure 6b.

| | AEP | | | |
|:---:|:---:|:---:|:---:|:---:|
| **Gage Extent (Model)** | $10^{-1}$ | $10^{-2}$ | $10^{-3}$ | $10^{-4}$ |
| TMSP01 | 5.06 | 8.23 | 12.83 | 19.81 |
| TMSP02 | 5.06 | 8.20 | 12.90 | 20.21 |
| TMSP03 | 4.95 | 7.88 | 12.16 | 17.77 |
| TMSP04 | 4.84 | 7.71 | 11.88 | 17.33 |

Plots of the extremal coefficient function versus normalized distance were shown for the three WRB and four TRB models in Figure 11. For both sets of models, the plots of the extremal coefficient function versus normalized distance varied (Figure 11a,c), with the models developed for the more limited gage extents exhibiting stronger extremal dependence as a function of distance. For each WRB/TRB model, pairwise sample site distances were calculated from the normalized coordinate values for the 85/218 study area simulation targets (Figure 5c/Figure 6b). The lower and upper bounds for these distances were tabulated and plotted (Figure 11b,d). For example, for the WRB models WMSP02, WMSP03,

and WMSP04, the lower and upper bounds were calculated to be 0.012/0.007/0.024 and 0.641/0.314/1.301, respectively. For the TRB models TMSP01, TMSP02, TMSP03, and TMSP04, the lower and upper bounds were calculated to be 0.0129/0.0095/0.0058/0.0043 and 2.99/2.15/1.31/0.98, respectively. From Figure 11b,d, one can observe that the range of extremal coefficient function values across the three WRB models and the four TRB models were rather consistent, underscoring that inter-site extremal dependence was modeled quite similarly within each set of models. The range of the extremal coefficient function values was greater for the four TRB models than it was for the three WRB models (Figure 11b,d). Extremal dependence was stronger for the three-day duration cool season (October to April) maxima throughout the 536 square kilometer Cougar dam study area in the WRB than it was for the 48 h MLC precipitation annual maxima in the 15,662 square kilometer TRB study area (Figure 11).

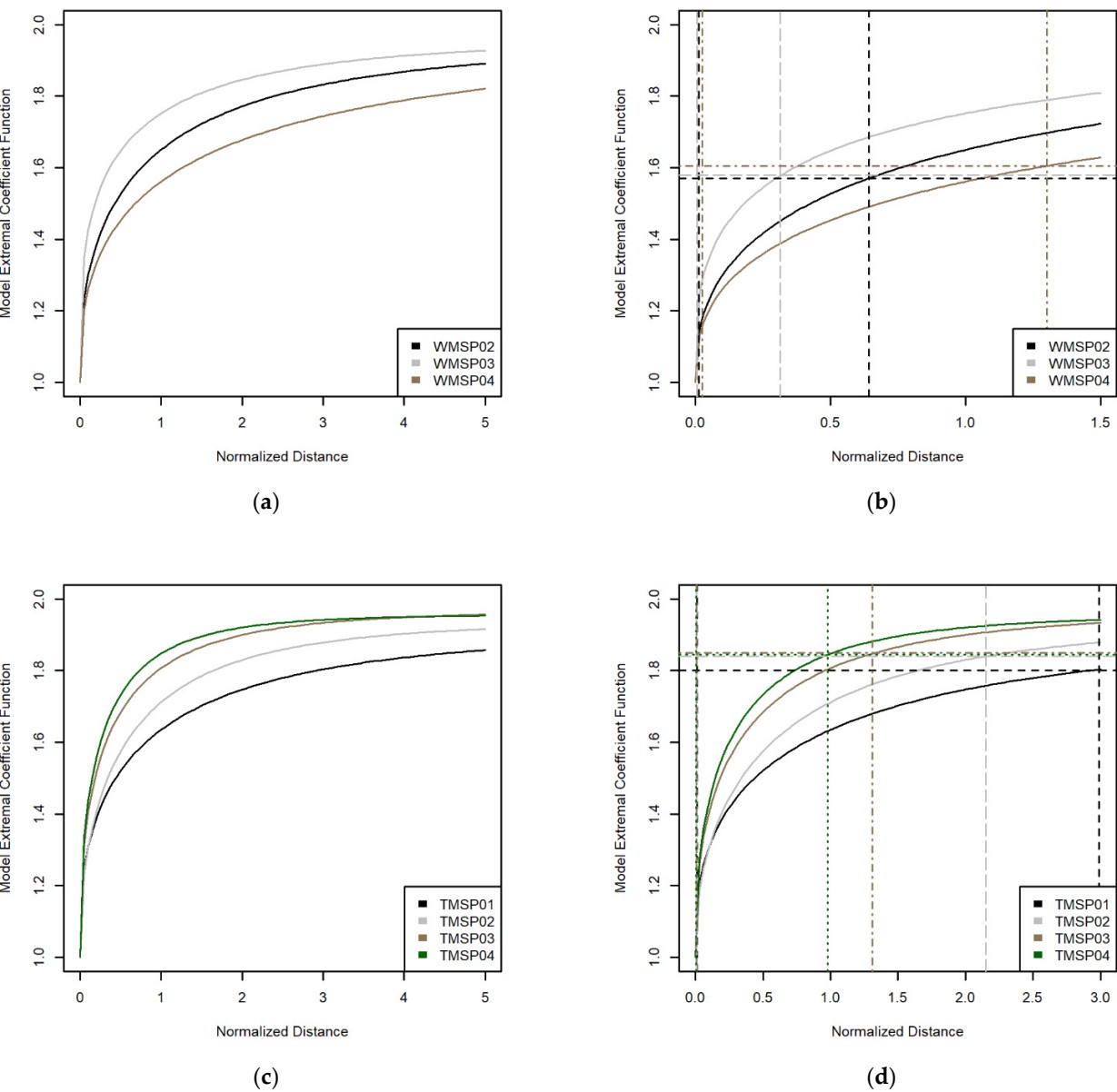

**Figure 11.** Plots of the extremal coefficient function versus normalized distance for the three WRB (**a**,**b**) and four TRB models (**c**,**d**). The lower and upper bounds of pairwise sample site distances calculated from the normalized coordinate values for the 85/218 study area simulation targets (Figure 5c/Figure 6b) are also plotted as vertical lines, with corresponding horizontal lines also included where the vertical lines intersect their respective model extremal coefficient function.

Tables [7] and [8] list three statistics, the coefficient of determination ($R^2$), Nash–Sutcliffe efficiency (NSE) [84], and Kling–Gupta efficiency (KGE) [85], that summarizes the degree of agreement among the gridded GEV location parameter, GEV scale parameter, and $10^{-3}$ AEP pointwise return level values computed from three WRB models and the four TRB models, respectively. Nash–Sutcliffe efficiency values range from minus infinity to one. An NSE value of one indicates a perfect match between the model and its observations. An NSE value of zero indicates that model predictions are as accurate as the mean of the observed data. NSE values less than zero indicate that the mean of the observed data is a better predictor than the model. Kling–Gupta efficiency values range from minus infinity to one. A model is more accurate when its KGE value is closer to one.

**Table 7.** The coefficient of determination ($R^2$), Nash–Sutcliffe efficiency (NSE) [84], and Kling–Gupta efficiency (KGE) [85] values summarizing the degree of agreement among the gridded GEV location parameter, GEV scale parameter, and $10^{-3}$ AEP pointwise return level values computed from the three WRB models (02 = WMSP02; 03 = WMSP03; 04 = WMSP04) for the Cougar dam project study area. The statistics were computed using the complete set of 853 grid cells that compose the project area at the base 30 arc-second grid cell resolution.

| | GEV Location | | | GEV Scale | | | Return Levels | | |
|---|---|---|---|---|---|---|---|---|---|
| | 02/03 | 02/04 | 03/04 | 02/03 | 02/04 | 03/04 | 02/03 | 02/04 | 03/04 |
| $R^2$ | 1.00 | 0.99 | 0.99 | 0.93 | 0.97 | 0.81 | 0.97 | 0.98 | 0.92 |
| NSE | 0.92 | 0.98 | 0.97 | 0.91 | 0.89 | 0.78 | 0.95 | 0.96 | 0.86 |
| KGE | 0.78 | 0.96 | 0.85 | 0.96 | 0.74 | 0.71 | 0.92 | 0.87 | 0.80 |

**Table 8.** The coefficient of determination ($R^2$), Nash–Sutcliffe efficiency (NSE) [84], and Kling–Gupta efficiency (KGE) [85] values summarizing the degree of agreement among the gridded GEV location parameter, GEV scale parameter, and $10^{-3}$ AEP pointwise return level values computed from the four TRB models (01 = TMSP02; 02 = TMSP02; 03 = TMSP03; 04 = TMSP04) for the TRB project study area. The statistics were computed using the complete set of 21,794 grid cells that compose the project area at the base 30 arc-second grid cell resolution.

| | GEV Location | | | | | GEV Scale | | | | | Return Levels | | | | |
|---|---|---|---|---|---|---|---|---|---|---|---|---|---|---|---|
| | 01/02 | 01/03 | 01/04 | 02/04 | 03/04 | 01/02 | 01/03 | 01/04 | 02/04 | 03/04 | 01/02 | 01/03 | 01/04 | 02/04 | 03/04 |
| $R^2$ | 0.98 | 0.98 | 0.98 | 0.99 | 1.00 | 0.99 | 0.92 | 0.85 | 0.88 | 0.85 | 0.99 | 0.94 | 0.89 | 0.93 | 0.90 |
| NSE | 0.96 | 0.97 | 0.83 | 0.72 | 0.85 | 0.93 | 0.81 | −2.18 | −1.01 | −0.87 | 0.98 | −1.75 | −5.92 | −5.53 | 0.62 |
| KGE | 0.94 | 0.91 | 0.83 | 0.89 | 0.93 | 0.86 | 0.81 | 0.29 | 0.50 | 0.55 | 0.88 | 0.73 | 0.34 | 0.51 | 0.68 |

The marginal distributions of a max-stable process are GEV distributed, possibly varying by location. Across each set of models, the GEV location and scale parameters were modeled more consistently throughout the Cougar Dam study area in the WRB than they were for the TRB study area (Tables [7] and [8]). Notably, for the four TRB models, while agreement among paired GEV scale parameter model grids for the TRB study area improved with increasing gage extent, the agreement was low. Table [9] lists the fitted values, including an estimate of their uncertainty, for the GEV shape parameter for the three WRB models and the four TRB models. For each set of models, the uncertainty of the GEV shape parameter estimate decreased with increased gage extent. The fitted values for the GEV shape parameter were more variable across the three WRB models than they were among the four TRB models. The remaining fitted values for the GEV shape parameter were 1.63 and 1.97 times the minimum GEV shape parameter value among the three WRB models whereas for the four TRB models, these values were 1.06, 1.17, and 1.18. Across the three WRB models, the order of the areal exceedances listed in Table [5] for the $10^{-3}$ and $10^{-4}$ AEP

aligned with the order of the fitted GEV shape parameter values reported for these three models in Table 9.

**Table 9.** Fitted values, including an estimate of their uncertainty, for the GEV shape parameter for the three Willamette River Basin models and the four Trinity River Basin models that were used to explore the impact of max-stable process areal exceedance calculations to precipitation gage extent.

| | **GEV Shape Parameter** | |
|---|---|---|
| **Model Name** | **Fitted Value** | **Standard Error** |
| WMSP02 | 0.02496 | 0.004246 |
| WMSP03 | 0.03027 | 0.003833 |
| WMSP04 | 0.01533 | 0.006999 |
| TMSP01 | 0.1441 | 0.009562 |
| TMSP02 | 0.146 | 0.007281 |
| TMSP03 | 0.1233 | 0.004677 |
| TMSP04 | 0.1306 | 0.003947 |

The computed pointwise return levels encapsulate modeling of the marginal distributions. For each set of models, as the gage extent increased, the agreement among the paired models' computed return levels increased, as measured by the NSE and KGE values reported in Tables 7 and 8. Interestingly, replacing the GEV scale grids for TMSP01, TMSP02, and TMSP03 with the GEV scale grid for TMSP04 resulted in the KGE statistic values increasing from 0.34, 0.51, and 0.68 (Table 9) to 0.98, 0.91, and 0.98 for the model pairs TMSP01/TMSP02, TMSP01/TMSP03, and TMSP01/TMSP04, respectively. Moreover, areal exceedance estimates for TMSP01, TMSP02, and TMSP03 changed from the values reported in Table 6 to 4.94/7.94/12.29/18.87, 4.97/7.97/12.46/19.47, and 4.86/7.67/11.77/17.10 for AEPs of $10^{-1}/10^{-2}/10^{-3}/10^{-4}$, respectively.

More accurately estimating the GEV scale parameter may improve agreement among the four TRB models such that based on the two storm types and model domains considered (i.e., three-day duration cool season maxima in the WRB of Oregon and 48 h mid-latitude cyclone precipitation annual maxima in Texas), in general, the impact the selected precipitation gage extent would have on areal exceedance estimates could broadly be reduced to estimation of the GEV shape parameter, particularly for rare AEP values. Two potential avenues to explore to improve GEV scale parameter estimation include the consideration of additional covariates or overfitting the scale model when applying the elastic-net penalty during trend surface development for model deployments with limited gage extent. While the uncertainty of the GEV shape parameter estimates decreased with increased gage extent, it is the uncertainty of the areal exceedances that are of most interest. Additional study is also needed to examine the impacts of gage extent on the uncertainty quantification of MSP modeled areal exceedances.

The set of twenty-six precipitation gages located within the one-degree-by-one-degree box with north, south, east, and west extents of 44.75° N, 43.75° N, −121.8° W, and −122.8° W (Figure 3c) that were used to develop WMSP04 using synthetic observations generated from the spatial process WMSP02 observed WMSP02 quite well based on the model comparisons and their reported degree of the agreement provided in Tables 5, 7 and 9 and Figure 11. A comparison of uncertainty estimates for the areal exceedance calculations from WMSP02 and WMSP04 is needed.

### 3.3. Impact of MSP Areal Exceedance Calculations to Precipitation Gage Density

Surface networks are generally not designed and maintained to observe on a regular grid. Within a given MSP model domain, some areas that require areal exceedance calculations may observe precipitation with high density while other areas may be data sparse. This analysis briefly explored the impact of gage density on MSP areal exceedance calculations.

For both the WRB and TRB, three subsets were randomly sampled from a base precipitation gage network. For the WRB, the base network was the 140 gages from Skahill et al. [49] with at least 20 seasonal maxima observations [50] that were located within the WRB and a 20-km buffer of the WRB watershed boundary (Figure 3a). The total number of gages used to develop the WRB models WMSP01, WMSP05, WMSP06, and WMSP07 were 140, 35, 70, and 105, respectively. For models WMSP01, WMSP05, WMSP06, and WMSP07, the number of precipitation gages within a 1-degree-by-1-degree box containing the WRB Cougar dam study area with north, south, east, and west extents of 44.5° N, 43.5° N, −121.6° W, and −122.6° W, were 23, 7, 10, and 12, respectively. For the TRB, the base network was the 610 gages located within a 3-degree buffer of the project study area (Figure 4a). The total number of gages used to develop the TRB models TMSP04, TMSP05, TMSP06, and TMSP07 were 610, 153, 305, and 458, respectively. For models TMSP04, TMSP05, TMSP06, and TMSP07, the number of precipitation gages within a 3.3 square degree box containing the TRB study area with north, south, east, and west extents of 33.8° N, 32.3° N, −96.6° W, and −98.8° W, were 63, 12, 29, and 47, respectively.

The areal exceedance estimates listed in Table 10 were obtained from one million independent copies for each spatial process, using their respective fitted values, for the 536 square kilometer Cougar Dam project study area using the sampling density shown in Figure 5b. The areal exceedance estimates listed in Table 11 were obtained from one million independent copies for each spatial process, using their respective fitted values, for the 15,662 square kilometer TRB project study area using the sampling density shown in Figure 6b.

**Table 10.** Areal exceedance estimates, in inches, obtained from one million independent copies for each spatial process, using their respective fitted values, for the 536 square kilometer Cougar Dam project study area using the sampling density shown in Figure 5b.

| | AEP | | | |
|---|---|---|---|---|
| **Gage Extent (Model)** | $10^{-1}$ | $10^{-2}$ | $10^{-3}$ | $10^{-4}$ |
| WMSP01 | 9.74 | 13.44 | 17.22 | 21.23 |
| WMSP05 | 9.64 | 13.73 | 18.26 | 23.35 |
| WMSP06 | 9.55 | 13.35 | 17.40 | 22.01 |
| WMSP07 | 9.70 | 13.47 | 17.36 | 21.06 |

**Table 11.** Areal exceedance estimates, in inches, obtained from one million independent copies for each spatial process, using their respective fitted values, for the 15,662 square kilometer Trinity River Basin project study area using the sampling density shown in Figure 6b.

| | AEP | | | |
|---|---|---|---|---|
| **Gage Extent (Model)** | $10^{-1}$ | $10^{-2}$ | $10^{-3}$ | $10^{-4}$ |
| TMSP04 | 4.84 | 7.71 | 11.88 | 17.33 |
| TMSP05 | 4.87 | 7.83 | 12.22 | 18.33 |
| TMSP06 | 4.91 | 7.89 | 12.35 | 18.74 |
| TMSP07 | 4.85 | 7.73 | 11.85 | 17.24 |

For the four models within each set, extremal dependence was modeled in a highly similar manner as measured by the extremal coefficient function plots in Figure 12. Also, for both sets of models, the agreement was good, as measured by the $R^2$, NSE, and KGE values reported in Tables 12 and 13, among the paired GEV location and scale parameter and pointwise return level model grids for each respective study area. The results in Tables 12 and 13 underscore the robustness of the novel approach that was employed for trend surface development as part of the deployment process for each MSP model [37].

The trend surface modeling methodology employed in this study addressed a previously mentioned potential drawback for MSP applications; viz., that it can be difficult to find accurate trend surfaces for the marginal parameters [11]. Across the four TRB models, there were slight differences in the GEV shape parameter estimates, including their uncertainty (Table 14). There were greater differences in the GEV shape parameter estimates among the WRB models. For both the WRB and TRB models, the order of the areal exceedances listed in Tables 10 and 11 for the $10^{-4}$ AEP for each study area aligned with the order of the fitted GEV shape parameter values reported for these models in Table 14. Additional study is needed to examine the impacts of gage density on the uncertainty quantification of MSP modeled areal exceedances.

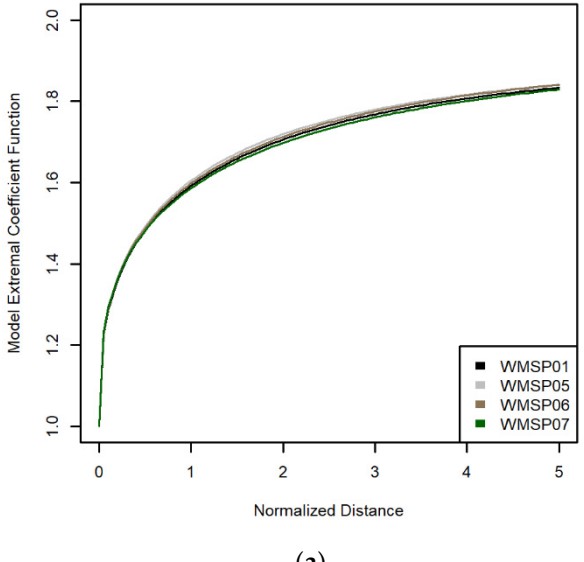

(**a**)

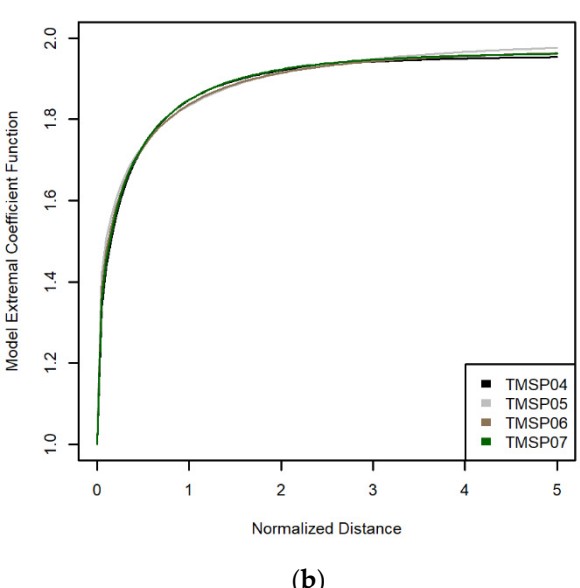

(**b**)

**Figure 12.** Plots of the extremal coefficient function as a function of normalized distance for each of the four models used to evaluate the impact of max-stable process areal exceedance calculations to precipitation gage density plots in the (**a**) Willamette River Basin and (**b**) Trinity River Basin.

**Table 12.** The coefficient of determination ($R^2$), Nash–Sutcliffe efficiency (NSE) [84], and Kling–Gupta efficiency (KGE) [85] values summarizing the degree of agreement among the gridded GEV location parameter, GEV scale parameter, and $10^{-3}$ AEP pointwise return level values computed from the four WRB models (01 = WMSP01; 05 = WMSP05; 06 = WMSP06; 07 = WMSP07) for the Cougar dam project study area. The statistics were computed using the complete set of 853 grid cells that compose the project area at the base 30 arc-second grid cell resolution.

| | GEV Location | | | | | GEV Scale | | | | | Return Levels | | | | |
|---|---|---|---|---|---|---|---|---|---|---|---|---|---|---|---|
| | 01/05 | 01/06 | 01/07 | 05/07 | 06/07 | 01/05 | 01/06 | 01/07 | 05/07 | 06/07 | 01/05 | 01/06 | 01/07 | 05/07 | 06/07 |
| **R²** | 0.94 | 1.00 | 1.00 | 0.93 | 1.00 | 0.98 | 0.99 | 0.99 | 0.99 | 1.00 | 0.96 | 1.00 | 1.00 | 0.98 | 1.00 |
| **NSE** | 0.64 | 0.96 | 0.91 | 0.91 | 0.99 | 0.84 | 0.81 | 0.90 | 0.81 | 0.49 | 0.43 | 0.99 | 0.99 | 0.70 | 0.99 |
| **KGE** | 0.57 | 0.84 | 0.76 | 0.87 | 0.93 | 0.63 | 0.95 | 0.95 | 0.70 | 0.92 | 0.68 | 0.98 | 0.92 | 0.81 | 0.94 |

*3.4. Impact of MSP Areal Exceedance Calculations on Model Fitting Method*

General MSP fitting involves the simultaneous estimation of trend surface and dependence model parameters. The optimization starting point for each general MSP fit was the vector of values obtained from its respective simple MSP calibration and fitted spatial GEV model. Four of the fourteen general MSP fits used constrained local optimization whereas the remaining ten models used unconstrained local optimization (Tables 1 and 15). Notably, the final GEV shape parameter estimate was always greater than its initial value across all fourteen general MSP fits (Table 15).

**Table 13.** The coefficient of determination ($R^2$), Nash–Sutcliffe efficiency (NSE) [84], and Kling–Gupta efficiency (KGE) [85] values summarizing the degree of agreement among the gridded GEV location parameter, GEV scale parameter, and $10^{-3}$ AEP pointwise return level values computed from the four TRB models (04 = TMSP04; 05 = TMSP05; 06 = TMSP06; 07 = TMSP07) for the TRB project study area. The statistics were computed using the complete set of 21,794 grid cells that compose the project area at the base 30 arc-second grid cell resolution.

| | GEV Location | | | | | GEV Scale | | | | | Return Levels | | | | |
|---|---|---|---|---|---|---|---|---|---|---|---|---|---|---|---|
| | 04/05 | 04/06 | 04/07 | 05/07 | 06/07 | 04/05 | 04/06 | 04/07 | 05/07 | 06/07 | 04/05 | 04/06 | 04/07 | 05/07 | 06/07 |
| $R^2$ | 0.91 | 0.98 | 1.00 | 0.94 | 0.99 | 0.91 | 0.95 | 0.97 | 0.90 | 0.98 | 0.96 | 0.96 | 0.98 | 0.96 | 0.99 |
| NSE | 0.85 | 0.95 | 0.99 | 0.85 | 0.97 | 0.72 | 0.61 | 0.86 | 0.88 | 0.86 | 0.77 | 0.37 | 0.97 | 0.82 | 0.21 |
| KGE | 0.82 | 0.91 | 0.96 | 0.83 | 0.94 | 0.90 | 0.75 | 0.87 | 0.80 | 0.83 | 0.97 | 0.76 | 0.90 | 0.88 | 0.82 |

**Table 14.** Fitted values, including an estimate of their uncertainty, for the GEV shape parameter for the four Willamette River Basin models and the four Trinity River Basin models that were used to explore the impact of max-stable process areal exceedance calculations on precipitation gage density.

| | GEV Shape Parameter | |
|---|---|---|
| Model Name | Fitted Value | Standard Error |
| WMSP01 | 0.008987 | 0.004033 |
| WMSP05 | 0.04064 | 0.006227 |
| WMSP06 | 0.03001 | 0.004287 |
| WMSP07 | 0.005042 | 0.004189 |
| TMSP04 | 0.1306 | 0.003947 |
| TMSP05 | 0.1315 | 0.004215 |
| TMSP06 | 0.1349 | 0.004135 |
| TMSP07 | 0.1273 | 0.003911 |

The only difference between the models WMSP01 and WMSP02 was the fitting method. Each MSP used the same extreme precipitation dataset, trend surface parameterization, dependence model, and set of initial values. WMSP01 used constrained local optimization whereas WMSP02 used unconstrained local optimization. Their final GEV shape parameter estimates differed by a multiplicative factor of approximately three. For WMSP01 and WMSP02 the ratios of their final to initial GEV shape parameter estimates were 1.32 and 3.68, respectively. The plot of the extremal coefficient function versus normalized distance, together with their initial estimate, is shown for each model in Figure 13. The final dependence parameter estimates for WMSP02 underestimate extremal dependence compared with its corresponding simple MSP fit. Interestingly, the calculated log-likelihood for WMSP02 of −2,552,281 was greater than its calculated value of −2,553,786 for WMSP01.

These differences among the two MSPs, in the final GEV shape parameter and dependence model parameter estimates, impact areal exceedance calculations for the Cougar dam safety study area as shown in Figure 14. The two plots in Figure 14 were generated using ten million simulations from each fitted process. The two areal exceedance curves begin to diverge for AEP values approximately less than $10^{-2}$. For AEP values of interest to dam and levee safety, their divergence has the potential to be notable. For example, the three-day basin average probable maximum precipitation estimate for Cougar Dam is 29.9 inches. From the fitted model for WMSP01, that equates to an AEP of $6.01 \times 10^{-7}$ whereas for WMSP02, the AEP estimate is $1.83 \times 10^{-6}$. These results underscored the known difficulty with fitting a general MSP model, particularly when a non-trivial marginal trend surface is desired [13,34,40,86].

**Table 15.** Model fitting method (C = constrained local optimization; U = unconstrained local optimization), initial values, and final fitted GEV shape parameter values for all fourteen max-stable models. The starting point for each general MSP optimization was the vector of values obtained from its respective simple MSP calibration and fitted spatial GEV model.

| Model Name | Model Fitting Method | GEV Shape Parameter | |
| --- | --- | --- | --- |
| | | Initial Estimate | Final Estimate |
| WMSP01 | C | 0.006785831 | 0.008987 |
| WMSP02 | U | 0.006785831 | 0.02496 |
| WMSP03 | U | 0.02036444 | 0.03027 |
| WMSP04 | U | $-0.00153159$ | 0.01533 |
| WMSP05 | C | 0.03778892 | 0.04064 |
| WMSP06 | C | 0.02769452 | 0.03001 |
| WMSP07 | C | 0.00277155 | 0.005042 |
| TMSP01 | U | 0.1138974 | 0.1441 |
| TMSP02 | U | 0.1246995 | 0.146 |
| TMSP03 | U | 0.1091662 | 0.1233 |
| TMSP04 | U | 0.117473 | 0.1306 |
| TMSP05 | U | 0.1192562 | 0.1315 |
| TMSP06 | U | 0.1201912 | 0.1349 |
| TMSP07 | U | 0.1155369 | 0.1273 |

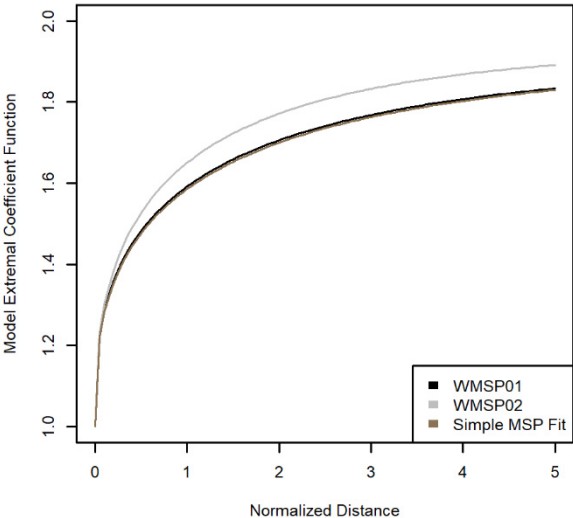

**Figure 13.** The plot of the fitted extremal coefficient function versus normalized distance for the max-stable models WMSP01 and WMSP02, together with their initial estimate, the simple MSP fit. The only difference between the models WMSP01 and WMSP02 was the fitting method.

Fourteen additional general MSP fits for WMSP01 were performed using constrained local optimization. For each fit, the specified box constraints were progressively relaxed. Figure 15 summarizes results from the fifteen fits for WMSP01 that were each obtained using constrained local optimization. For each model fit, it plots its deviation from the preferred model, its initial value, versus its negative log-likelihood. The curve defined by the fifteen points in Figure 15 clearly demonstrated the trade-off between model and data fit when using composite likelihoods [29]. The trade-off curve can be used for general MSP model selection.

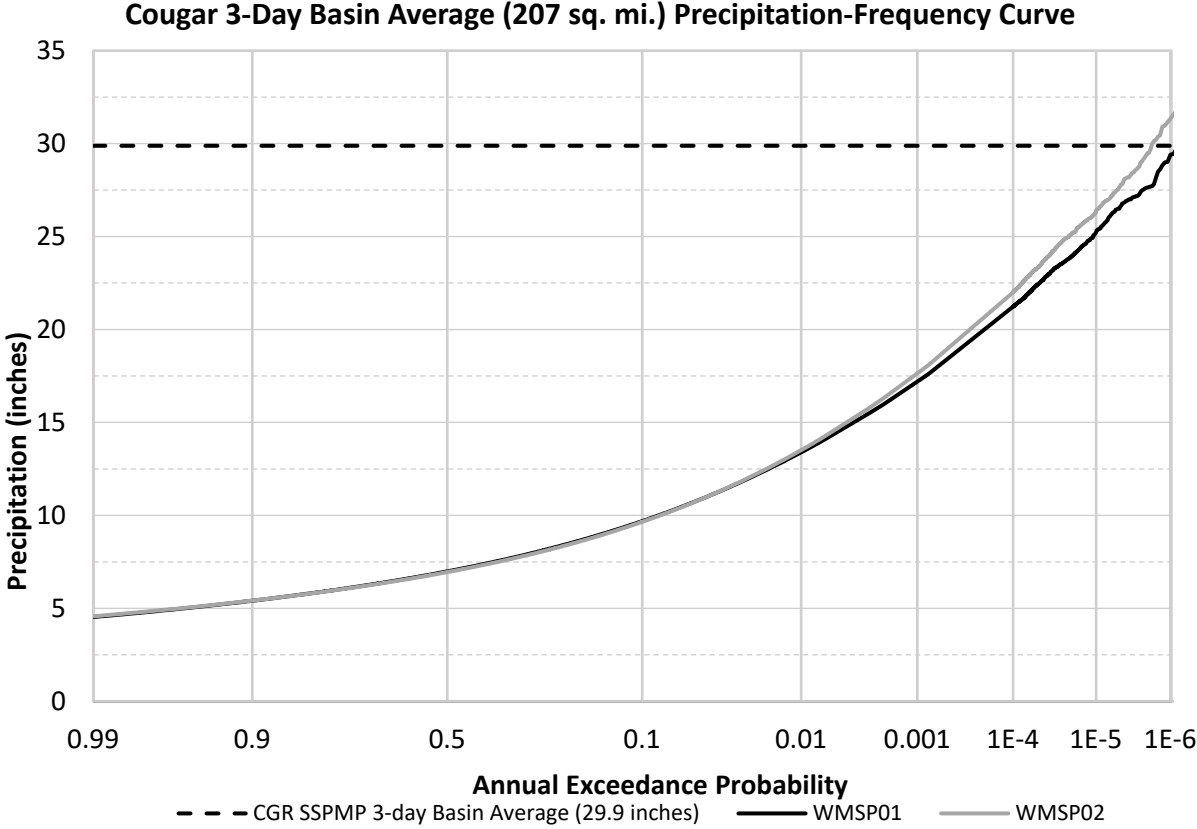

**Figure 14.** Areal exceedance calculations for the Cougar Dam safety study area, were generated using 10,000,000 simulations using the fitted values from the process (WMSP01 and WMSP02). The only difference between the models WMSP01 and WMSP02 was the fitting method. The three-day basin average probable maximum precipitation estimate for Cougar Dam is 29.9 inches. From the fitted model for WMSP01, that equates to an AEP of $6.01 \times 10^{-7}$ whereas for WMSP02, the AEP estimate is $1.83 \times 10^{-6}$.

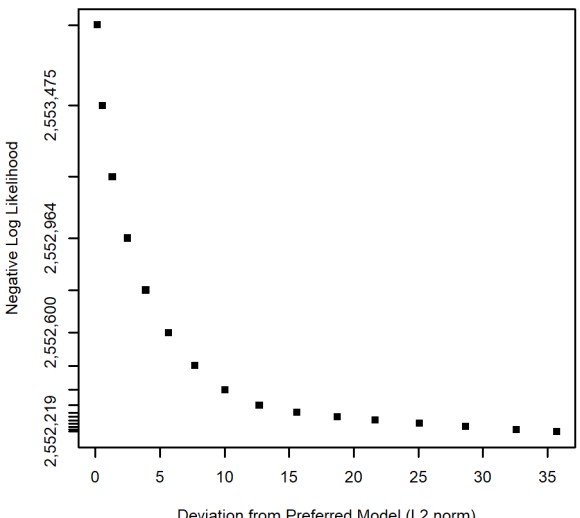

**Figure 15.** Results from fifteen fits for WMSP01 using constrained local optimization wherein the specified box constraints were progressively relaxed for each model fit. The only difference between each fit were the specified box constraints. The horizontal axis was defined to be the Euclidean distance between fitted and preferred model parameters where the preferred model was its specified initial value, i.e., the vector of values obtained from the model's respective simple MSP calibration

and fitted spatial GEV model. The vertical axis was defined as the negative log-likelihood value associated with each model fit. The trade-off curve defined by the plotted results obtained from the model fits can be used for general MSP model selection.

While efficiency is important [19], further evaluation of alternative fitting methods is needed; for example, a regularized inversion approach permits an analysis of the tradeoff between model and data fit [87]. Evaluating the impacts of the fitting method on areal exceedance uncertainty is also needed.

## 4. Conclusions

The application of an MSP model enables the estimation of areal-based exceedances within an EVT framework. It is inherently a spatial model, does not require a decomposition of a study area into homogeneous regions or the introduction of uncertain empirical areal reduction factors for computing areal exceedances, and it has a strong and coherent mathematical basis for model fitting, selection, extrapolation, and uncertainty quantification.

As MSP models become more broadly used for areal precipitation-frequency analysis, guidance and lessons learned regarding their practical application are needed. This study explored the impacts of max-stable process areal exceedance calculations to study area sampling density, surface network precipitation gage extent, model fitting method, and gage density, for which the first three are user specified for any given model deployment. Two distinct extreme storm types from two separate geographical locations were used for model development and application.

The explorations directed at study area sampling density showed that MSPs can be efficiently and dynamically deployed to support dam and levee safety applications, including uncertainty quantification.

The potential impacts of the selected model fitting method on areal exceedance calculations, particularly those relevant to the dam and levee safety, were shown to be non-trivial. However, a curve that defines the trade-off between data and preferred model fit can guide the selection of the general MSP to be used for model application.

Several opportunities were identified for future related applied research, including, among others, further study of methods to reduce study area sampling density for efficient MSP application, the evaluation of additional covariates to improve GEV scale parameter estimation for modeling MLCs in Texas, evaluations that also include uncertainty quantification of the areal exceedance calculations and the application of general MSP fitting methods that penalize deviations from a preferred parameter state.

**Author Contributions:** Conceptualization, B.S., C.H.S., B.T.R. and J.F.E.; methodology, B.S., C.H.S., B.T.R. and J.F.E.; formal analysis, B.S.; investigation, B.S., C.H.S., B.T.R. and J.F.E.; writing—original draft preparation, B.S.; writing—review and editing, B.S., C.H.S., B.T.R. and J.F.E.; funding acquisition, B.S., C.H.S. and J.F.E. All authors have read and agreed to the published version of the manuscript.

**Funding:** This research was funded by the U.S. Army Corps of Engineers Risk Management Center. The APC was funded by *Hydrology*.

**Institutional Review Board Statement:** Not applicable.

**Informed Consent Statement:** Not applicable.

**Data Availability Statement:** The data presented in this study are available on request from the corresponding author.

**Acknowledgments:** The authors thank the two reviewers for their comments which improved this article.

**Conflicts of Interest:** The authors declare no conflict of interest.

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
