# Peer review of "Impacts of Max-Stable Process Areal Exceedance Calculations to Study Area Sampling Density, Surface Network Precipitation Gage Extent and Density, and Model Fitting Method"

_hydrology, doi:10.3390/hydrology10060121_

Round 1

Reviewer 1 Report

Reviewer compliments authors on their thorough investigation as the analysis of areal exceedance is of great importance for hydrological modelling. However, the authors should consider the following remarks in their next revision.

# General remarks
1. Conclusions section should provide a short-listed answers to the four questions raised in the Introduction with a what-to-do-next takeaway.
2. Authors might want to consider making data available through their institution hosting service or through a free hosting service like zenodo.

#Specific remarks
1. Figure 2, colours for entries TX and Project are similar. It would be more readable if one of these changes to e.g. red.
2. Figure 3, subfigs a, c-f show different gage subsets over a topography. As markers are rather thin, at first readers cannot notice the difference in layout. Authors should increase the size of gage markers (or make them more significant in any other way).
3. Figure 5, some elements are not understandable on a first reading. In subfig a. location of the "Cougar dam contributing area" is black. On the other hand, the same colour is used in subfigs b-d but the polygon is shaded in white. Authors should try to use colours in more-or-less the same way in one figure (otherwise it might confuse the readers).
4. In figures 5 and 6, it seems that the first entry in legend shares the same colour with second entry. Authors should use some different colours for distinct entries in the legend (and in plots as well).

Author Response

Dear Hydrology Editorial and Reviewer Boards,

We are submitting our revised article titled “Impacts of Max-Stable Process Areal Exceedance Calculations to Study Area Sampling Density, Surface Network Precipitation Gage Extent and Density, and Model Fitting Method” for peer-review for publication in Hydrology. We confirm that neither the manuscript nor any parts of its content are currently under consideration or published in another journal.

Directly below we explain, point by point, the details of the revisions to the manuscript and our responses to the referees’ comments.

We look forward to hearing back from you regarding your decision.

Thank you.

Sincerely,

Brian Skahill

Haden Smith

Brook Russell

John England

Replies to comments for reviewer 1

First, a sincere thank you for reviewing our manuscript. Your review comments were helpful in strengthening the manuscript and we acknowledge your contribution in the Acknowledgments section of the article.

Comment 1 (Reviewer compliments authors on their thorough investigation as the analysis of areal exceedance is of great importance for hydrological modelling. However, the authors should consider the following remarks in their next revision.)

Thank you.

Comment 2 (Conclusions section should provide a short-listed answers to the four questions raised in the Introduction with a what-to-do-next takeaway.)

The Conclusions section was slightly altered to address your comment. Thank you.

Comment 3 (Authors might want to consider making data available through their institution hosting service or through a free hosting service like zenodo.)

We decided to keep our data availability statement the same --- study data is available upon request from the corresponding author.

Comment 4 (Figure 2, colours for entries TX and Project are similar. It would be more readable if one of these changes to e.g. red.)

Noted, corrected. Thank you.

Comment 5 (Figure 3, subfigs a, c-f show different gage subsets over a topography. As markers are rather thin, at first readers cannot notice the difference in layout. Authors should increase the size of gage markers (or make them more significant in any other way).)

Noted, corrected. Thank you.

Comment 6 (Figure 5, some elements are not understandable on a first reading. In subfig a. location of the "Cougar dam contributing area" is black. On the other hand, the same colour is used in subfigs b-d but the polygon is shaded in white. Authors should try to use colours in more-or-less the same way in one figure (otherwise it might confuse the readers).)

Noted, corrected. Thank you.

Comment 7 (In figures 5 and 6, it seems that the first entry in legend shares the same colour with second entry. Authors should use some different colours for distinct entries in the legend (and in plots as well).)

Noted. Thank you.

Reviewer 2 Report

The research is very interesting and comprehensive, however, there are flaws to be resolved.

Line 37 – 38, This sentence is not enough to introduce the topic and to understand the usefulness of estimates of extreme precipitation levels. I recommend adding introductory sentences in relation to the usefulness for both flood hazard and hydrogeological risk (flood hazard could be homogenised in the wording as hydraulic risk). We recommend that research aimed at the mitigation of hydrogeological and hydraulic risks, in which the analysis of extreme precipitation events plays a predominant role, be reported in the text:

Vasu, N.N., Lee, S.R., Pradhan, A.M.S., Kim, Y.T., Kang, S.H. and Lee, D.H., 2016. A new approach to temporal modelling for landslide hazard assessment using an extreme rainfall induced-landslide index. Engineering Geology, 215, pp.36-49.

Gentilucci, M., Materazzi, M., & Pambianchi, G. (2021). Statistical Analysis of Landslide Susceptibility, Macerata Province (Central Italy). Hydrology, 8(1), 5.

Madsen, H., Lawrence, D., Lang, M., Martinkova, M. and Kjeldsen, T.R., 2014. Review of trend analysis and climate change projections of extreme precipitation and floods in Europe. Journal of Hydrology, 519, pp.3634-3650.

Line 57-60, put the citations in square brackets without making that list of authors and modifying the sentence.

Line 72-99, avoid questions in the text to summarise the main and innovative points of your research, formulating an organic speech.

Line 102, the mile is not a measure in the international system, change to Km (replace it throughout the rest of the paper).

Figure 1, Improve the definition and on figure 1b, insert the hillshade in transparency below the dem.

Figure 2, Impreve the definition.

move Figure 1 and Figure 2 and insert them after their respective citations in the text.

all figures should be consistent in terms of the legend, in relation to altitude, figure 3b differs from the others please make the legend the same and include for all of them a hillshade in transparency as for figure 1.

The discussion assessing the results in light of the most recent literature on the subject is completely missing.

Author Response

Dear Hydrology Editorial and Reviewer Boards,

We are submitting our revised article titled “Impacts of Max-Stable Process Areal Exceedance Calculations to Study Area Sampling Density, Surface Network Precipitation Gage Extent and Density, and Model Fitting Method” for peer-review for publication in Hydrology. We confirm that neither the manuscript nor any parts of its content are currently under consideration or published in another journal.

Directly below we explain, point by point, the details of the revisions to the manuscript and our responses to the referees’ comments.

We look forward to hearing back from you regarding your decision.

Thank you.

Sincerely,

Brian Skahill

Haden Smith

Brook Russell

John England

Replies to comments for reviewer 2

First, a sincere thank you for reviewing our manuscript. Your review comments were helpful in strengthening the manuscript and we acknowledge your contribution in the Acknowledgments section of the article.

Comment 1 (The research is very interesting and comprehensive, however, there are flaws to be resolved.)

Thank you.

Comment 2 (Line 37 – 38, This sentence is not enough to introduce the topic and to understand the usefulness of estimates of extreme precipitation levels. I recommend adding introductory sentences in relation to the usefulness for both flood hazard and hydrogeological risk (flood hazard could be homogenised in the wording as hydraulic risk). We recommend that research aimed at the mitigation of hydrogeological and hydraulic risks, in which the analysis of extreme precipitation events plays a predominant role, be reported in the text:

Vasu, N.N., Lee, S.R., Pradhan, A.M.S., Kim, Y.T., Kang, S.H. and Lee, D.H., 2016. A new approach to temporal modelling for landslide hazard assessment using an extreme rainfall induced-landslide index. Engineering Geology, 215, pp.36-49.

Gentilucci, M., Materazzi, M., & Pambianchi, G. (2021). Statistical Analysis of Landslide Susceptibility, Macerata Province (Central Italy). Hydrology, 8(1), 5.

Madsen, H., Lawrence, D., Lang, M., Martinkova, M. and Kjeldsen, T.R., 2014. Review of trend analysis and climate change projections of extreme precipitation and floods in Europe. Journal of Hydrology, 519, pp.3634-3650.)

Noted. The first paragraph of the Introduction was modified to address your comment(s). Thank you.

Comment 3 (Line 57-60, put the citations in square brackets without making that list of authors and modifying the sentence.)

Noted, corrected. Thank you.

Comment 4 (Line 72-99, avoid questions in the text to summarise the main and innovative points of your research, formulating an organic speech.)

Noted, corrected. Thank you.

Comment 5 (Line 102, the mile is not a measure in the international system, change to Km (replace it throughout the rest of the paper).)

Noted, corrected. Thank you.

Comment 6 (Figure 1, Improve the definition and on figure 1b, insert the hillshade in transparency below the dem.)

Noted, corrected. Thank you.

Comment 7 (Figure 2, Impreve the definition.)

Noted, corrected. Thank you.

Comment 8 (move Figure 1 and Figure 2 and insert them after their respective citations in the text.)

Noted. Thank you.

Comment 9 (all figures should be consistent in terms of the legend, in relation to altitude, figure 3b differs from the others please make the legend the same and include for all of them a hillshade in transparency as for figure 1.)

Noted, corrected. Thank you.

Comment 10 (The discussion assessing the results in light of the most recent literature on the subject is completely missing.)

Noted, corrected. Thank you.

Round 2

Reviewer 2 Report

The manuscript is sufficiently improved, please pay even more attention to the discussion.